# Synthesis of ARM User Facility Surface Rainfall Datasets to Construct a Best Estimate Value Added Product (PrecipBE)

Israel Silber[1], Jennifer M. Comstock[2], Adam K. Theisen[3], Michael R. Kieburtz[2], Zeen Zhu[4], Jenni Kyrouac[3]

[1] Atmospheric, Climate, and Earth Sciences Division, Pacific Northwest National Laboratory, Richland, WA, 99352, USA
[2] Advanced Computing, Mathematics, and Data Division, Pacific Northwest National Laboratory, Richland, WA, 99352, USA
[3] Environmental Science Division, Argonne National Laboratory, Lemont, IL 60439, USA
[4] Environmental Science and Technologies Department, Brookhaven National Laboratory, Upton, NY 11973, USA

*Correspondence to*: Israel Silber (israel.silber@pnnl.gov)

**Abstract.** Surface precipitation measurements are essential for Earth system model (ESM) evaluation and understanding cloud processes. An ever-growing need for robust, temporally evolving, and easy-to-use statistical datasets provides motivation for a baseline ground-based precipitation properties data product. The U.S. Department of Energy Atmospheric Radiation Measurement (ARM) user facility operates an extensive suite of precipitation instruments with various sensitivities and operating mechanisms, which render the decision of which instrument to use based on one or more fixed thresholds challenging

and prone to errors and bias. Using a long-term instrument inter-comparison from a unique per-precipitation event perspective, rather than instantaneous sample comparison, we demonstrate that ARM rainfall-measuring instruments are generally consistent with each other at the statistical level. Inter-instrument deviations at the single event level can be large, especially at specific rainfall event properties such as maximum precipitation rates. A machine-learning (ML) analysis using a random forest regressor indicates that in some cases, depending on instrument, local site climatology, and/or specific deployment

configuration, certain atmospheric state variables influence the measured quantities in an unpredictable manner. Thus, a-priori weighing of different instruments does not necessarily lead to more accurate and less biased synthesis of instrument data. These results motivate the design of the ARM precipitation best-estimate (PrecipBE) value-added product, which incorporates all valid precipitation data while considering data quality and other instrument limitations.

PrecipBE consists of time series and tabular statistics datasets in an easy-to-use and insightful per-precipitation event format.

It provides a large set of precipitation event properties supplemented with ancillary data from various ARM datasets that correspond to the detected precipitation events. We describe the PrecipBE algorithm and demonstrate its use via the examination of a single-day output as well as a long-term trend analysis of precipitation events at the ARM Southern Great Plains (SGP) site, covering more than 30 years of data. The trend analysis tentatively suggests a long-term tendency for mainly shorter and less intense precipitation events at the SGP site, but a long-term increase in annual rainfall by more than 36 mm

(5%) per decade. This rainfall trend is catalyzed primarily by more extreme event properties of relatively rare, intense precipitation events, with event total and 1-minute maximum precipitation rate at a 1-year timeframe increasing up to 5 mm and 9 mm/hr (several percent) per decade, respectively. While the currently available PrecipBE datasets (at https://adc.arm.gov/discovery/) cover rainfall from multiple ARM deployments up to March 2025, PrecipBE is planned to be

expanded to include solid-phase precipitation and will soon become an operational product with a several-day lag from real-

time. We invite the ARM user community to leverage this new product and welcome user feedback to further enhance the

dataset.

## 1 Introduction

Surface precipitation measurements serve as a crucial benchmark in Earth system model (ESM) evaluation (e.g., Emmenegger et al., 2022; Mikkelsen et al., 2024; Zhang et al., 2017) and aerosol-cloud interactions (ACI) studies (e.g., Christensen et al., 2024; Martin et al., 2017), among other process understanding efforts. Detailed case studies using surface precipitation data often require temporally evolving precipitation rate and accumulation data to account for the dynamic nature and short time scales of cloud evolution relative to the typically slower-evolving atmospheric state (e.g., Bretherton et al., 2010). These time series data serve as target quantities (benchmarks) for model simulations or analytical models. Certain precipitation-characterizing disdrometers, such as laser and video disdrometers, provide additional observational constraints on the precipitation properties, such as hydrometeor particle size distributions (PSDs). ESM evaluation studies, on the other hand, often rely on bulk statistics or data subsets and, therefore, utilize isolated precipitation event statistics after conditioning on quantities such as surface temperature, for example.

The U.S. Department of Energy Atmospheric Radiation Measurement (ARM) user facility (Mather, 2024; Mather et al., 2016) operates multiple types of precipitation-measuring instruments, including impact (Bartholomew, 2016a), video (Bartholomew, 2020b), and laser disdrometers (Bartholomew, 2020a), as well as tipping and weighing bucket rain gauges (Bartholomew, 2019; Kyrouac and Tuftedal, 2024). Each instrument tends to have higher sensitivity and/or better accuracy at certain precipitation conditions (e.g., Ciach, 2003; Fehlmann et al., 2020; Ro et al., 2024; Wang et al., 2021). For example, the Pluvio2 weighing bucket operated by ARM tends to be robust at high rainfall rates (Ro et al., 2024; Saha et al., 2021). The OTT Parsivel2 (LDIS; Bartholomew, 2020a), distributed in many ARM sites, is generally considered robust, but has been shown to suffer from biases at a specific drop size range (e.g., Raupach and Berne, 2015) and to underestimate the vertical velocity of drops larger than 1 mm, which translates to precipitation rate underestimation (Tokay et al., 2014). Similarly, the two-dimensional video disdrometer (VDIS; Bartholomew, 2020b) is often treated as a reference precipitation instrument, specifically when the drop PSDs are of interest (e.g., Tokay et al., 2020). However, this instrument is more likely to underestimate rainfall amounts in cases with drops smaller than roughly 0.3 mm (corresponding to its first size bin) or when large drops (> ~2.4 mm; often commensurate with heavy precipitation) are observed, due to terminal velocity underestimation (e.g., Tokay et al., 2013).

The availability of independent studies evaluating the performance of precipitation instruments under strict laboratory conditions (e.g., Colli et al., 2013; Lanza et al., 2010; Lanza and Vuerich, 2009; Saha et al., 2021) is still scarce. Moreover, comprehensive analyses of precipitation errors as a function of various background conditions (high wind, etc.) and deployment configurations (e.g., Montero-Martínez et al., 2016; Montero-Martínez and García-García, 2016; Wang et al., 2021), let alone snowy conditions (e.g., Battaglia et al., 2010; Milewska et al., 2019; Yuter et al., 2006), is still limited and requires additional research. In the interim, however, determining the "*true*" precipitation properties or weighing different ARM instrument samples based on the current literature is prone to unpredictable errors and biases. Therefore, as comprehensively discussed below, straightforward statistics combining data from measurements collected (per deployment)

would ostensibly provide the best estimates of precipitation event properties (onset and ending, accumulation, precipitation rates, etc.).

Here, we first present a long-term multi-instrument inter-comparison of rainfall event data collected at the ARM Southern Great Plains (SGP; Sisterson et al., 2016) observatory (Section 2). Supported by the application of a machine learning (ML) algorithm (a random forest regressor), this analysis underscores the challenge in such cases of multi-instrument data without a clear and consistent *"true"* benchmark. The results from this comparison serve as a strong motivation for a best-estimate data product implementing straightforward statistics. These comparison results are also used to guide the design of the ARM precipitation best-estimate (PrecipBE) value-added product (VAP), the processing algorithm of which is elaborated on in Section 3. Section 4 describes PrecipBE's data structure, and Section 5 presents a brief trend analysis using more than 30 years of ARM precipitation data from the ARM SGP site, available on the ARM Data Discovery (https://adc.arm.gov/discovery/). Conclusions and a short outlook are given in Section 6.

## 2 Instrument Inter-Comparison as Motivation for a Best-Estimate Data Product

### 2.1 Data Processing

Which precipitation instrument has the most reliable precipitation readings and should be used by default in given conditions? An answer to this question is not trivial. First, precipitation instruments have different sensitivities, which are influenced by ambient conditions and are often impacted by the same variables they aim to measure, namely, precipitation amount, rate, or particle properties, as noted above. In addition, those instruments have minimum quantization sizes, which could result in inconsistencies concerning precipitation event onset and ending times, leading to differences in event totals. As such, data mining efforts aimed at determining those instrument strengths and weaknesses require a baseline definition of precipitation events instead of typical instantaneous sample comparisons. In this section, we perform an inter-comparison on a per-event basis by examining inter-instrument differences in rainfall event properties.

The analysis focuses on rainfall data collected at the ARM SGP site's co-located central (C1) and extended facility 13 (E13) over a 14-year period, from January 10, 2011, to January 10, 2025. A list of the instruments and data products analyzed is provided in Table 1. (Refer to https://armgov.svcs.arm.gov/capabilities/observatories/sgp for site information and central facility layout.) For a given instrument, we define a rainfall event as a set of accumulated precipitation samples (at temperatures greater than 3 °C) with gaps between neighboring precipitation readings (samples) shorter than 30 min. (larger gaps in event definition such as 60 min were tested and exhibited minor changes; not shown). Instrument events continuing to the next day are concatenated as long as they follow the same 30-min maximum gap logic. If the total accumulation in a given instrument event is smaller than 0.1 mm, it is omitted from this analysis. Instrument events that failed quality control (QC) checks (for calibration issues, bad samples, etc.) in some or all event samples are also omitted from this analysis. Finally, a given event is also omitted if it indicates highly unlikely statistics; specifically, event total > 300 mm, event period > 5 days, mean precipitation rate > 120 mm/hr, and/or 1-min average maximum precipitation rate > 300 mm/hr. Some of these thresholds have

been met and confirmed in recorded history (e.g., Koralegedara et al., 2019; Lagouvardos et al., 2013), but to our knowledge, have not previously occurred during ARM deployments. However, these thresholds are rarely exceeded in instrument samples, for various reasons, and account for up to a few percent (< 2.5%) of precipitation events detected using all ARM SGP

instruments (counting from 2011), except for the optical rain gauge (ORG; Bartholomew, 2016b), with nearly 9% of detected events having one or more variables exceeding these thresholds. We note that ARM is in the process of retiring the ORG, which will not serve as a data source going forward.

**Table 1: Precipitation instruments and data products included in the analysis presented in Section 2 and incorporated in the**
**PrecipBE value-added product. The effective quantization increments refer to the reported precipitation variable's increments converted to mm/min.**

| Abbreviated name | Description | Temporal resolution and effective quantization increments | Reference |
|---|---|---|---|
| PWD[1,2] | Vaisala RAINCAP acoustic sensor as part of the Present Weather Detector, a component of the surface meteorological system (MET) at the main observatory | 1 min<br>0.01 mm/min | (Kyrouac et al., 2021; Kyrouac and Tuftedal, 2024) |
| AOSMET[1,2] | Vaisala RAINCAP acoustic sensor as part of the meteorological station associated with the Aerosol Observing System (~10-meters above ground) | 1 sec<br>0.00016 mm/min | (Kyrouac, 2019a; Kyrouac and Tuftedal, 2010) |
| DISDROMETER[1,2,3] | Joss-Waldvogel impact disdrometer | 1 min<br>0.00001 mm/min | (Bartholomew, 2016a; Wang, 2006) |
| VDISQUANTS[1,2] | Joanneum Research two-dimensional video disdrometer quantities value-added product | 1 min<br>0.00006 mm/min | (Bartholomew, 2020b; Hardin et al., 2020, 2021) |
| LDQUANTS[1,2] | OTT Parsivel2 laser disdrometer quantities value-added product | 1 min<br>0.00006 mm/min | (Bartholomew, 2020a; Hardin et al., 2020, 2021) |
| WBPLUVIO2[1,2] | OTT Pluvio2 weighing bucket rain gauge | 1 min<br>0.01 mm/min | (Bartholomew, 2019; Zhu et al., 2016) |
| TBRG[1,2] | Novalynx Tipping bucket rain gauge; commonly part of the MET system | 1 min<br>0.256 mm/min | (Kyrouac et al., 2021; Kyrouac and Tuftedal, 2024) |

| METWXT, PRECIPMET, MARINEMET, and ABMMET[2] | Vaisala RAINCAP acoustic sensor as part of the Vaisala WXT520 or WXT530 meteorological instrument systems installed at various ARM and ARM-related facilities | 1 sec<br>0.00016 mm/min | (Holdridge and Kyrouac, 2012; Howie et al., 2016; Kyrouac, 2019b; Kyrouac et al., 2017; Kyrouac and Shi, 2018; Reynolds et al., 2017) |
|---|---|---|---|
| PWS[2,4] | Vaisala FD12P Present Weather Sensor meteorological system | 1 min; 0.00016 mm/min | (Kyrouac and Tuftedal, 2001; Ritsche, 2008) |
| RAINWB[1,4] | Belfort weighing bucket rain gauge | 5 min; 0.001 mm/min | (Bartholomew, 2016c; Shi et al., 2010) |
| ORG[1,4] | Optical Scientific, inc optical rain gauge; commonly part of the MET system | 1 min; 0.00015 mm/min | (Bartholomew, 2016b; Kyrouac et al., 2021) |

[1]Included in the analysis presented in Section 2.
[2]Incorporated in PrecipBE (where available)
[3]ARM changed the DISDROMETER code name to IDIS starting 2025-04-08, outside the date span examined in this study
[4]Retired instrument

To streamline the interpretation of analysis results, we select a "reference" instrument to examine deviations of events from one instrument to another. Thus, we inter-compare pairs of instruments, with one of them being the "reference" instrument. This "reference" instrument is not a *"true"* benchmark, as in the case of the World Meteorological Organisation (WMO) rainfall intensity intercomparison, for example (Lanza et al., 2010; Lanza and Vuerich, 2009; Vuerich et al., 2009), during which only maximum precipitation rates per event were evaluated against a reference set of carefully calibrated rain gauges. Here, however, the related biases of the "reference" instrument can still be characterized. For example, in cases where most or all other precipitation instruments show a consistent deviation from the reference, we can tentatively conclude that the observed bias originates in the reference instrument.

Ideally, the best reference instrument would be the tipping bucket rain gauge (TBRG), because it was the first deployed precipitation instrument at the ARM SGP site (since 1993), and is still operational, covering the whole operation period of all other precipitation instruments. However, the TBRG has a very coarse precipitation amount least count (minimum detection of 0.254 mm; 0.1 inch; cf. Table 1), rendering its sensitivity and general accuracy (in weak events) inadequate for serving as a reference instrument (as demonstrated below), especially compared to other instruments such as disdrometers. Therefore, we chose to use the Present Weather Detector (PWD), which is integrated in the ARM Surface Meteorological System (MET; Kyrouac and Tuftedal, 2024), as the reference instrument. The PWD has a very long record at the ARM SGP site, starting on January 10, 2011, enabling inter-comparison with a wide range of instruments.

One of the main challenges in a per precipitation-event multi-instrument inter-comparison is associating individual instrument precipitation events with the reference instrument event, primarily due to the different onset and event duration times. This

could explain why, to our knowledge, event characterization is typically limited to the synthesis of only two instruments (i.e., instrument pairs), a specific case that is more straightforward to resolve (e.g., Keefer et al., 2008), or operating on fixed-duration windows such as defining an event as a day with recorded precipitation above a certain set of thresholds as in the case of the WMO intercomparison (which in practice, also used the "instrument pairs" approach). This challenge is exemplified in the simplified diagram shown in Figure 1. In this case, three precipitation events are identified in the PWD data (reference

instrument). One or more events detected with other instrument data can be aggregated and become associated with a given reference instrument event (as a single event). For example, events 1 and 2 detected using the LDIS are associated with the PWD's event number 1, while events 3, 4, and 5 detected using the TBRG data are associated with the PWD's event number 2. However, to prevent event conflicts in the inter-comparison, multiple reference instrument events cannot be associated with a single event detected using a different instrument. In such cases, the instrument events are omitted from the inter-comparison.

For example, event 1 detected using the VDIS or the LDIS event 4. In the latter case, we have interlacing conditions, resulting in the exclusion of LDIS event 3 as well since including it would likely result in a negative bias when comparing it to the PWD's event number 2.

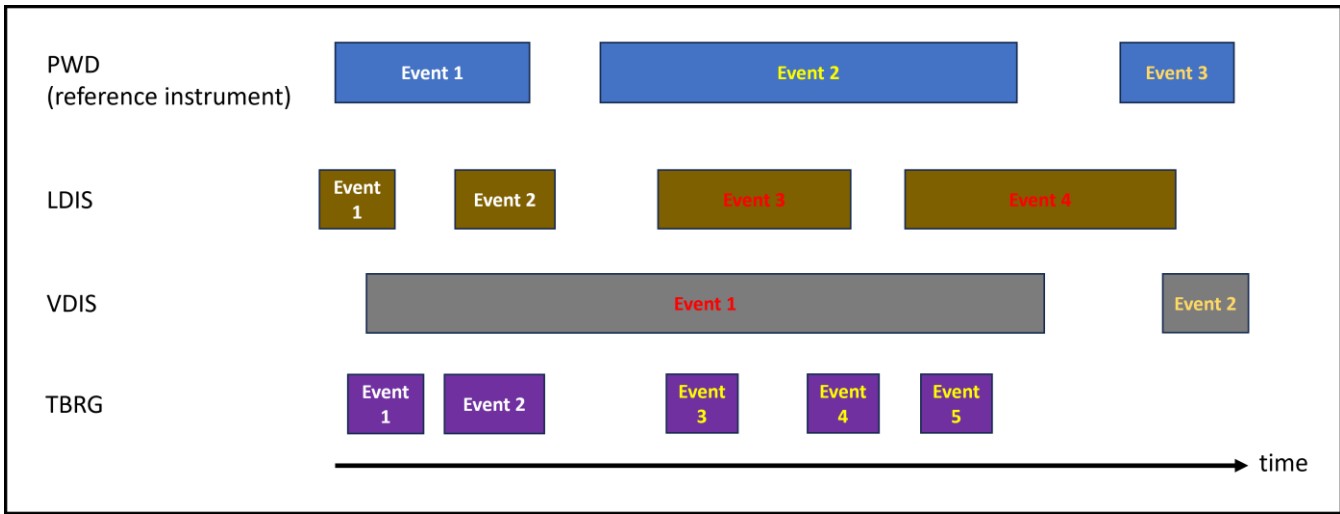

**Figure 1: Simplified diagram exemplifying the challenge of associating precipitation events detected using different instruments when a reference instrument is used. Here, the present weather detector (PWD) serves as the reference instrument, and its events are designated using different font colors. Precipitation events detected using the laser disdrometer (LDIS), the video disdrometer (VDIS), or the tipping bucket rain gauge (TBRG) become associated with PWD events only if they are not conflicting with it (event font colors match the associated PWD events). Conflicting events are designated using the red font color.**


This event association and aggregation exercise results in the removal of some instrument pair events. Removal percentages range from 0.7% of TBRG events to 53% of the PWD event pairs with the Belfort weighing bucket rain gauge (RAINWB). Smaller conflicting percentages, such as in the case of the TBRG or the Pluvio2 weighing bucket (WBPLUVIO2) with 4.5% of events being conflicted with the reference instrument, are often the result of the compared instrument tending to record

shorter events than the reference (see the TBRG versus PWD example in Figure 1). Larger conflicting percentages, such as in the case of the RAINWB or the Joss-Waldvogel impact disdrometer (DISDROMETER) data, with 47%, often occur when the compared instrument tends to longer events than the reference instrument (see the VDIS events versus PWD example in Figure 1). We note that the filtering of QC-flagged or anomalous reading events prior to the aggregation exercise had minor influence on analysis results (not shown), but it could theoretically be more impactful in other cases.

## 2.2 Inter-Comparison Results

Figure 2 shows probability density functions (PDFs) of precipitation (rainfall) event total amount based on the PWD (panel a) and event total deviations of different ARM instruments from the reference (i.e., the PWD; panels b-i). The distribution of event total amounts is strongly skewed (Figure 2a) with a PWD-estimated average of 5.3 mm, within the third distribution tercile. The three terciles are mapped to the deviation PDFs in panels b-i, and indicate that the smallest deviations tend to be

associated with the first tercile, whereas the largest deviations between instruments and the reference occur in top-tercile events, with deviations consistently being smaller than their associated terciles' right edge. Combined with the shape of the PDFs, it is suggested that the vast majority of ARM precipitation instruments tend to be consistent with each other, with mean deviations ($\mu$) smaller than 3 mm in magnitude and variability (represented here by the standard deviation; $\sigma$) being smaller than 10 mm. The PWD appears to be consistent to the greatest extent with the TBRG and the WBPLUVIO2 (means of 0.5 mm

or less; $\sigma$ on the order of 5 mm; in Figure 2b and Figure 2i, respectively). Some instruments and data products tend to record larger event totals relative to the PWD (e.g., LDQUANTS in Figure 2f, AOSMET in Figure 2g) whereas others exhibit a tendency for smaller totals (e.g., VDISQUANTS in Figure 2d, DISDROMETER in Figure 2h). These patterns are robust with the same qualitative results and minor quantitative variations if only events with totals greater than 1 mm are analyzed, for example, and deviations appear directly susceptible only to the magnitude of the evaluated variable (i.e., event total) in the

reference instrument, as indicated by the mapped terciles (and examined via linear regression; not shown). While the RAINWB is statistically consistent on average with the PWD (Figure 2e), its variability is somewhat greater than the other instruments. However, it is the ORG's deviations that stand out with a much larger variability (~12 mm) and an average overestimation by more than 4 mm (Figure 2c) (see also Kyrouac and Tuftedal, 2024). This overestimation becomes stark when conditioning on event totals greater than 1 mm with an average deviation from the reference of +8 mm.

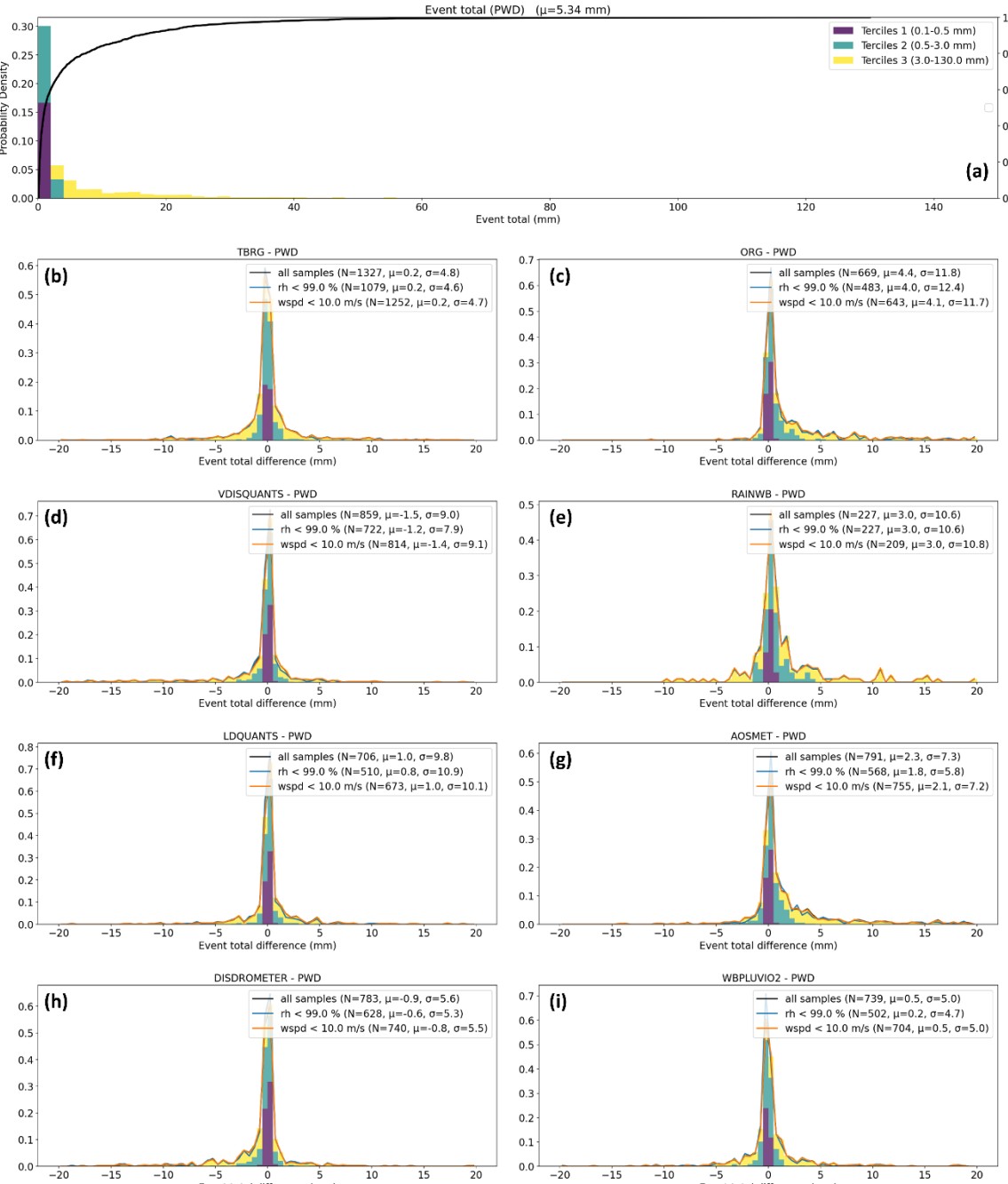


Figure 2: (a) Probability density function (PDF) of ARM SGP precipitation (rainfall) event total amount based on the PWD (bin width of 1.0 mm) and (b-i) PDFs of instrument deviations from the PWD, serving in this inter-comparison as the reference instrument (bin width of 0.5 mm). The purple, green, and yellow colored bars denote the three terciles of the PWD data (see legend in panel a), which are mapped to the histograms in panels b-i. The blue and orange curves designate histograms calculated while

conditioning on event-mean relative humidity (omitting likely foggy conditions) and wind speed (omitting strong winds), respectively, both derived from MET observations. In each panel, see the legends for the total number of event samples ($N$), mean deviation ($\mu$), and standard deviation ($\sigma$). Legend quantity units are mm.

The differences between instrument precipitation event measurements are relatively more variable when examining event periods (Figure 3). Similar to the event total, event periods are positively skewed (Figure 3a), averaging at 105 min, just above

the second tercile. The ORG measurements suggest precipitation events that are even more strongly skewed than the PWD, with durations longer by more than 40 min, on average, and considerable relative inconsistency (deviation $\sigma$ exceeding 130 min; Figure 3c). (Note that some of the positive PDF skewness is influenced by the aggregation and filtering methodology discussed above). The DISDROMETER shows a greater tendency, with precipitation events lasting 120 min longer on average, and a trend toward extreme values in cases within the third tercile (yellow-shaded areas in Figure 3h). The RAINWB exhibits

an even more substantial positive bias, exceeding 6 hours (Figure 3e). These long-event tendencies reflect the challenge in aggregating precipitation events, which resulted in the exclusion of a large subset of samples taken by those instruments from this analysis. In fact, the RAINWB event period bias and errors are so large, to an extent that is highly challenging to reconcile in an integrated dataset without introducing significant biases. In this regard, the PWD role as a reference instrument can be justified in the current analysis by the instrument's precipitation measurement properties being "somewhere in the middle"

across the ARM precipitation instrument suite. The PWD's event period statistics and general instrument behavior is in good agreement with the VDISQUANTS and LDQUANTS VAPs (Figure 3d and Figure 3f, respectively), with average deviations of a few minutes, as well as with the AOSMET with average deviations of 12 min (Figure 3g). The TBRG (Figure 3b) and WBPLUVIO2 (Figure 3i) display negatively skewed deviation distributions, with mirror-like patterns compared to the ORG and DISDROMETER, with some TBRG events lasting a few minutes, all the while the corresponding PWD events exceeding

1 hour (see the second tercile's edge in Figure 3b).

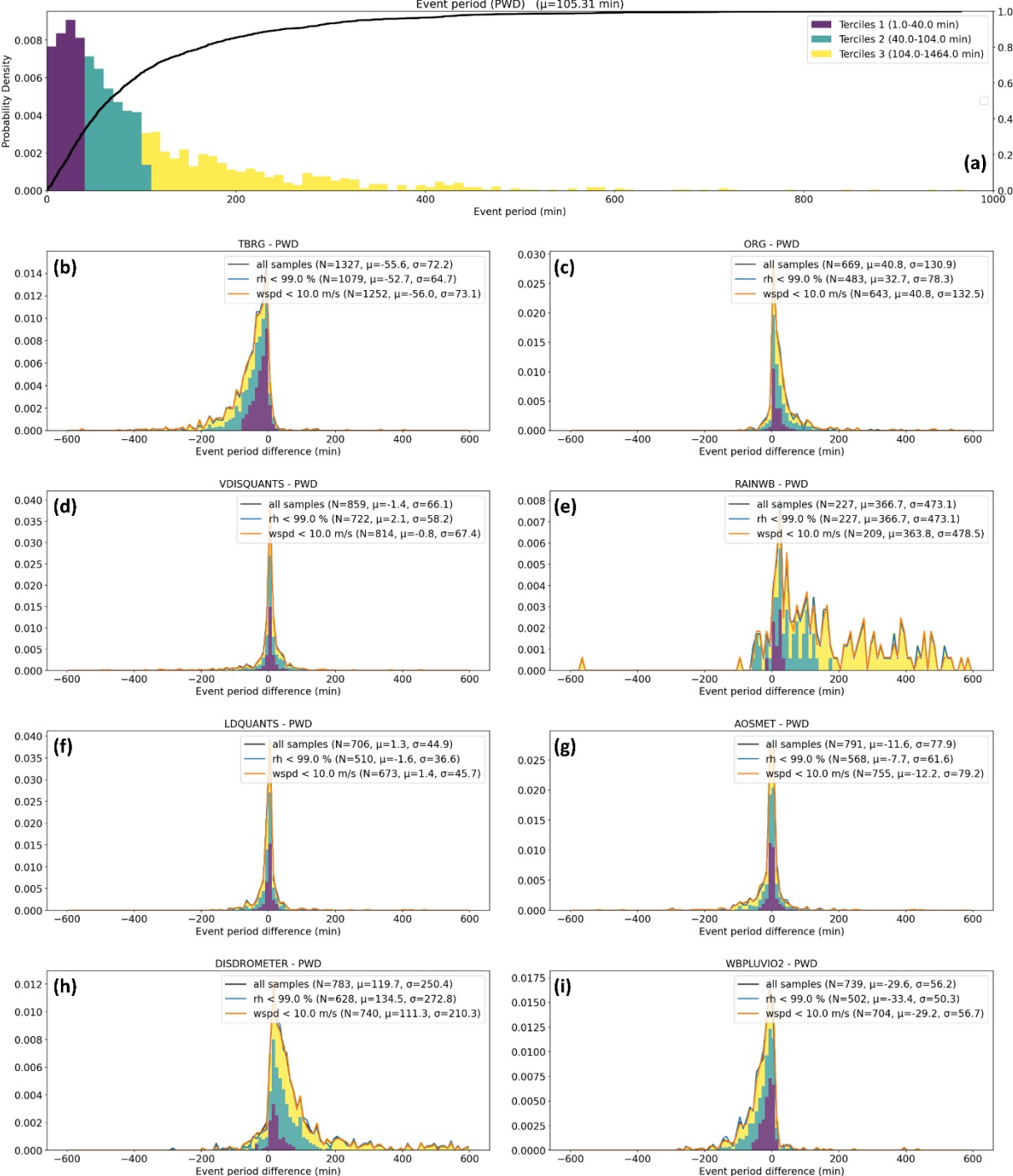

Figure 3: As in Figure 2, but for the event period (bin widths of 10 min; legend quantities are given in units of min).

The event 1-min-average maximum and event-mean precipitation rate comparisons (Figure 4 and Figure 5, respectively) suggest that most instruments are generally consistent with each other at the bulk level, especially in the case of mean rates, with all instruments except for the TBRG having average differences from the reference smaller than 2 mm/h (0.4 mm/h or less in the case of the VDISQUANTS and LDQUANTS; Figure 5d and Figure 5f, respectively). All instruments except for the TBRG and ORG also exhibit standard deviations of 4 mm/h or less.

We do not see any indications for a significant negative bias of the event maximum precipitation rate by the PWD or a positive bias by the LDQUANTS product (Parsivel2), as suggested by the WMO intercomparison (cf. Lanza et al., 2010). Single-event deviations can be quite large, as depicted by distribution tails, but from a bulk perspective, the instrument pair deviations generally tend to be evenly distributed around 0 mm/h in most cases, and the PWD is most consistent with the LDQUANTS (Figure 4f) and WBPLUVIO2 (Figure 4i). The agreement with the WBPLUVIO2 averages at 0.0 mm/h when conditioning on event totals greater than 1 mm and wind speed smaller than 10 m/s (not shown), close to the operation and filtering conditions of the WMO intercomparison (see Lanza and Vuerich, 2009; Vuerich et al., 2009; sect. 3.1). This result contrasts with the WMO intercomparison, where the Pluvio exhibited the highest performance (Lanza et al., 2010; their Table 2). This contrast is potentially influenced by deployment setup, site-specific factors, and/or sample size (the WMO intercomparison used approximately 1/10 the number of precipitation events analyzed here).

In both the mean and 1-min maximum precipitation rates, the TBRG (Figure 4b and Figure 5b) exhibits a distinct bi-modal PDF shape, which originates in its coarse minimum least count of 0.254 mm. The events associated with the secondary peak in the mean precipitation rate histogram (Figure 5b) are at the third PWD distribution tercile (yellow-shaded area), i.e., intense enough to be detected by the TBRG, but too weak and/or short to form consistent correspondence with the other instruments, and possibly influenced by some residual water on the bucket's "spoon". The coarse TBRG least count, combined with the 1-min sampling resolution, also results in weak events being below the TBRG's detection limit, as evident by the lack of first tercile events (purple-shaded areas) based on the maximum precipitation rate (Figure 4b) and very few weak events when partitioned based on mean event precipitation rate (Figure 5b). Accounting for this instrument limitation by omitting precipitation events with total amounts less than 1 mm results in behavior consistent with the aforementioned instruments and the disappearance of the bi-modal PDF artifact (not shown), suggesting that higher event total thresholds should be used for the TBRG in an integrated data product. The negative (positive) event period tendency of the TBRG in Figure 3b (DISDROMETER in Figure 3h) are compensated by the positive (negative) event-mean precipitation rates observed in Figure 5b (Figure 5h), resulting in a net event amount that is in agreement with other instruments, as indicated in Figure 2. While the 1-min maximum precipitation characteristic is generally the most variable across the various instruments (Figure 4) due to the irregular, potentially tempestuous nature of precipitation over the commonly-used 1-min precipitation instrument averaging period, combined with sensitivity limitations of different instruments, the ORG and RAINWB exhibit a much more erratic behavior. Specifically, The RAINWB significantly underestimates both the event 1-min maximum precipitation rate (Figure 4e) and event-mean precipitation rate (Figure 5e), which provides an extreme case of error compensation resulting in a moderate bias, as seen in the event total amount PDF (Figure 2e).

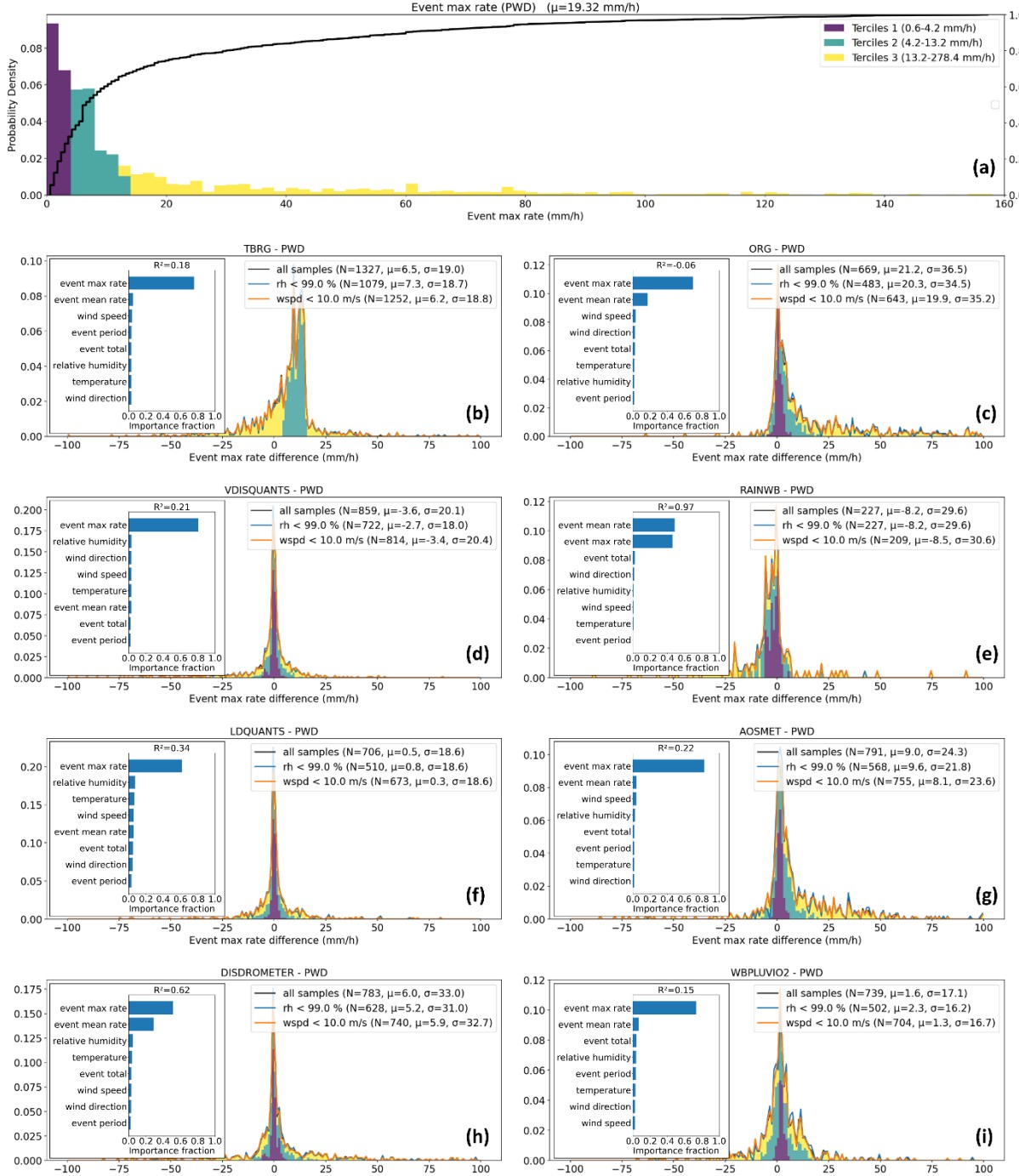

Figure 4: As in Figure 2, but for event 1-min-averaged maximum precipitation rate (bin widths of 2 and 1 mm/h in panels a and b-i, respectively; legend quantities are given in mm/h units). The inset panels show feature importance analysis of various PWD event properties and event-mean atmospheric state variables derived from MET observations. The feature importance results are derived from a random forest regression model fit (see text) with the coefficient of determination specified at the top of the inset.

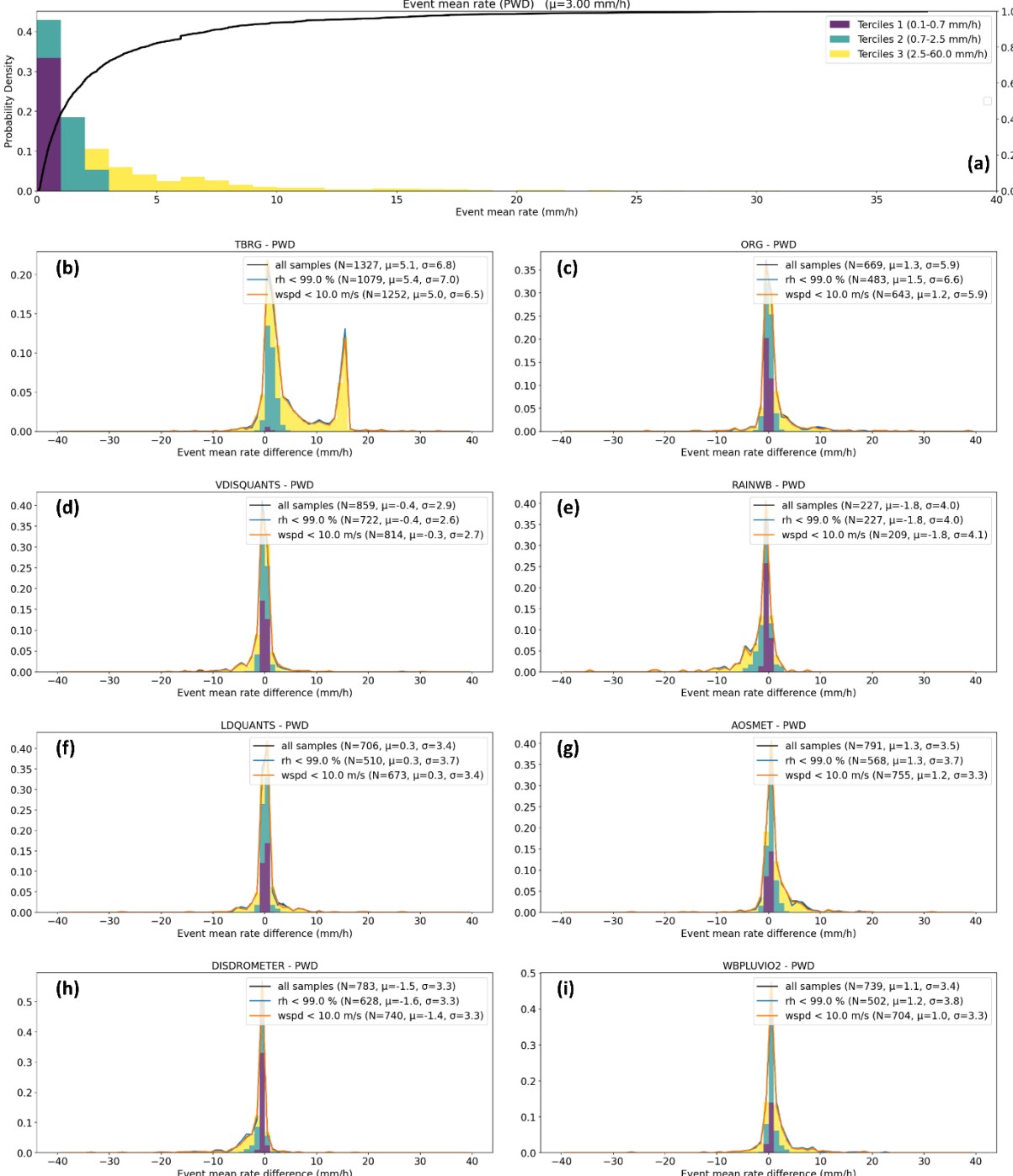

Figure 5: as in Figure 2 but for event-mean precipitation rate (in mm/h; bin widths of 1 mm/h).

### 2.3 Instrument Sensitivity, Deployment Configuration, and Atmospheric State Effects on the Inter-Comparison Results

Ambient conditions can influence deviations between instrument measurements, often as a function of the instrument operation mechanism (e.g., Bartholomew, 2016c, 2020a; Kyrouac and Tuftedal, 2024; Montero-Martínez et al., 2016; Wang et al., 2021). In the secondary (curved line) PDFs illustrated in Figure 2, Figure 3, Figure 4, and Figure 5, we examined the impact of some of these influencing factors. Specifically, by excluding events with event-mean relative humidity exceeding 99% (likely foggy conditions; accounting for the MET system's uncertainty) or events with event-mean wind speeds higher than 10 m/s (high winds), some of the potential impact of these forcings on event statistics can be evaluated. The effect of relative humidity (RH) appears somewhat limited, with mixed behavior of increasing or decreasing the deviation magnitude or standard deviations. This mixed and weak behavior could be due to the RH threshold used and/or because the examined instruments are less influenced by ambient moisture effects. The wind speed PDFs, however, indicate that conditioning for high winds tends to reduce the instrument deviation mean and standard deviations in all four examined parameters.

To further demonstrate the often site-dependent challenge of disentangling the influence of different parameters on differences in precipitation event measurements and statistics, we conduct a feature importance analysis using the Random Forest (RF) regressor in the Scikit-Learn Python package (Pedregosa et al., 2011). We also perform the analysis using datasets from two additional ARM deployments: The main site at Houston, Texas, of the Tracking Aerosol Convection Interactions Experiment (TRACER; Jensen et al., 2023) spanning October 1, 2021 through October 2, 2022, and the Eastern North Atlantic central site at Graciosa Island (ENA; see Mather, 2024; Wood et al., 2015) spanning October 1, 2013 through January 14, 2025, representing convective- and stratiform-dominated regimes, respectively. (From this aspect, the SGP site, often characterized by continental shallow convection, serves as a season-dependent mixture of the two regimes).

The feature importance analysis enables ranking the factors (features) that are most influential on the fitted RF model; i.e., features that have the most impact on the prediction of the model's target variable (in this case, inter-instrument deviations). Using the default algorithm's hyperparameters (100 estimators/trees, unlimited tree depth, etc.), we input as features the four precipitation event properties from the reference instrument (total, period, mean, and 1-min maximum precipitation rates) as well as the event-mean temperature, RH, pressure, wind speed, and wind direction measured by the MET system. We run the algorithm separately for instrument pairs and event properties; that is, a single RF algorithm run examines the deviations of an instrument pair across the four event properties (the target variable). Because the purpose of this ML exercise is qualitative, for brevity, we only present the results for the run using the event-maximum precipitation rate, depicted for the SGP, TRACER, and ENA sites in the insets shown in Figure 4, Figure 6, and Figure 7, respectively. We present, but overlook, RF fits in which the resultant coefficient of determination ($R^2$) is negative, indicating a fit with no predictive skill.

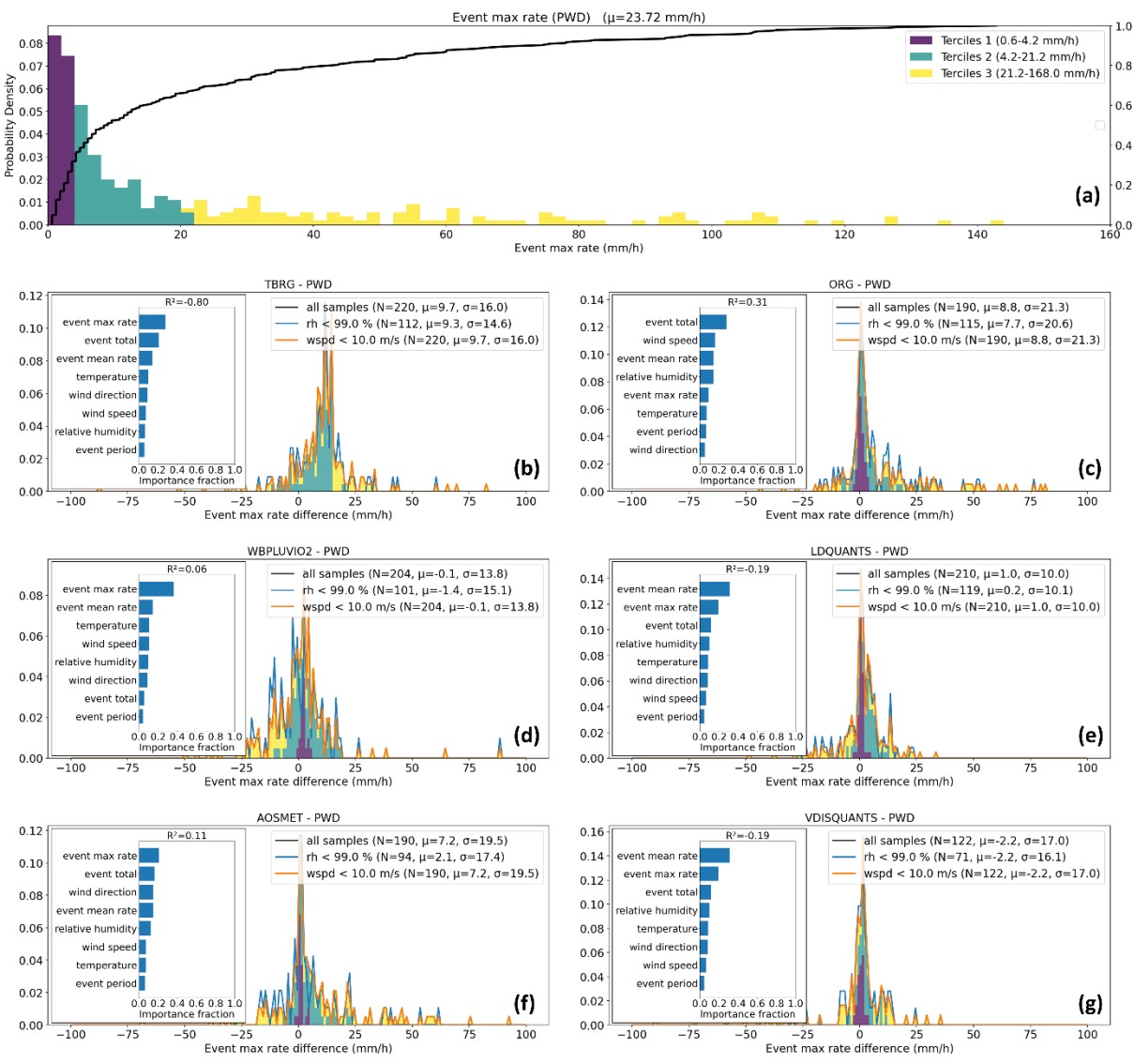

**Figure 6: as in Figure 4 but for the Tracking Aerosol Convection Interactions Experiment (TRACER) main site.**

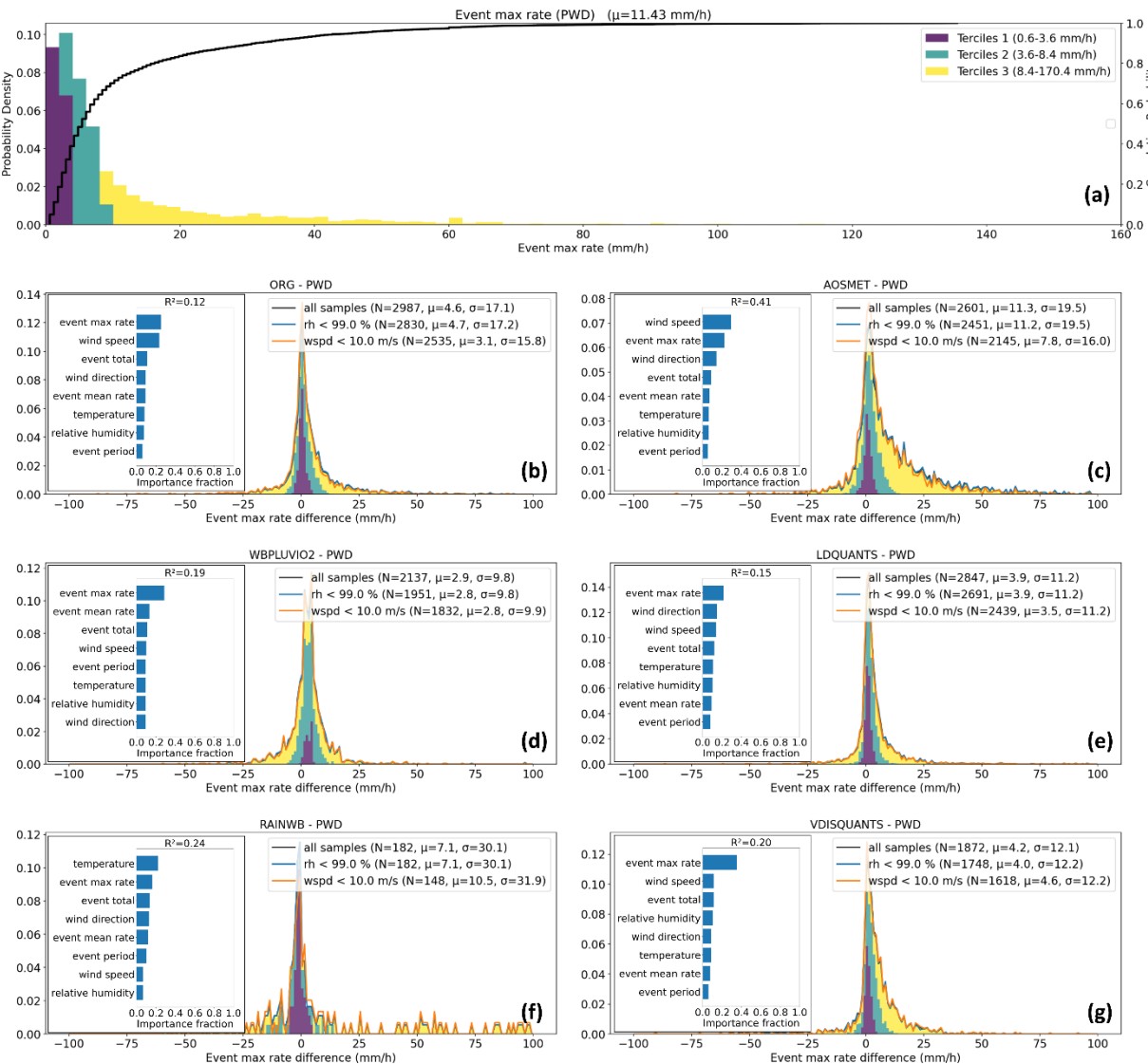

Figure 7: as in Figure 4 but for the ARM Eastern North Atlantic (ENA) central site.

The PWD's 1-min maximum precipitation rate distribution for the TRACER main site (Figure 6a) supports its designation as a "convective" site with an average value (23.7 mm/h) greater by roughly 20% than the SGP and second tercile values greater than the SGP site's value by nearly 40%. The ENA distribution, on the other hand, exhibits a tendency toward weaker instantaneous precipitation rates, around 40% lower than at the SGP site, based on mean values and second-tercile statistics in Figure 7a. However, the general patterns of instrument deviations relative to the reference (PWD in all cases) indicate similar tendencies with regard to the average deviation and variability, noting that the ENA PWD (Figure 7) tends to report lower maximum rates relative to all other instruments, which might be an indication of an instrument bias (cf. Lanza et al., 2010).

Focusing on the SGP feature importance results (Figure 4b-i), all instrument pairs with the PWD, except for the RAINWB-PWD pair, are most influenced by a large margin by the event-max precipitation rates, suggesting the existence of some proportionality between the deviations and the variable itself. This proportionality was also indicated in joint distributions we tested, and the general dominance of the examined target variable with its deviation feature (estimated relative errors for that matter) was seen in the vast majority of cases (not shown). In those analyses, the other examined features typically showed very weak, if any, proportionality (not shown). The RAINWB-PWD pair (Figure 4e) has the event-mean rate as the most dominant feature, which is also highly impactful in the case of the DISDROMETER (ranked 2nd; see Figure 4h), reflecting the tendency for some inconsistent behavior of those instruments, as discussed above.

The feature importance analysis of the 1-min maximum precipitation rates is less conclusive for the TRACER dataset, with the two most important features being one of the event maximum precipitation rate, the mean precipitation rate, and the event total (Figure 6b-g). This result is likely influenced by the convective nature of this site's dataset, reflecting the tendency of heavy precipitation events to be associated with large amounts, high intensity, and relatively short duration (hence, the mean rates are relatively high as well). However, in some cases, such as the wind speed in the AOSMET-PWD pair (Figure 6f), lower-ranked variables are comparable in amplitude, suggesting site- or deployment-specific constraints, some of which are not predictable in advance without a detailed analysis. The wind speed is also the most informative (ranked 1st) for the AOSMET-PWD pair in the ENA dataset (Figure 7c). This might suggest a wind-dependent AOSMET instrument bias, but could alternatively indicate a more general issue in the deployment configuration and/or the combination of site climatology and weaknesses of some instruments given that wind speed is ranked 2nd in the ORG-PWD and the VDISQUANTS-PWD pairs (Figure 7b and Figure 7g) and wind direction and speed are ranked 2nd and 3rd, respectively, in the LDQUANTS-PWD pair (Figure 7e). As noted earlier, the accuracy of those instruments is known to be susceptible to high winds; hence, they are likely more influenced by the climatologically stronger winds at the ENA site than at the TRACER site. (None of the TRACER events are associated with event-mean winds stronger than 10 m/s; compare the orange and black curve statistics in Figure 6b-g).

A tentative conclusion that can be drawn from these results is that weighing different instruments based on their evaluated sensitivities and accuracies from the literature can result in greater bias due to unmatching background conditions as well as unanticipated confounding factors, particularly when combining climatological factors with specific deployment setups. While additional quantitative characterization of instrument susceptibilities to deployment properties and conditions is essential, a deployment-dependent study of this type requires significant effort. The outcomes of such extensive efforts are highly challenging to predict in advance. Therefore, these data characterization studies often take place post-deployment, when the collected dataset is sufficiently large to produce substantial results (beyond the scope of this study).

Suppose one wishes to develop an operational, unbiased (or at least, bias-mitigated) precipitation best-estimate data product. In that case, given that they do not have a *true* benchmark, they need to be aware of all the factors described above by performing robust characterization, which would ideally require a best-estimate product — this presents a conundrum. A first step towards resolving this conundrum would be to assume, given the evidence from this inter-comparison about instrument

consistency, that the suite of ARM instruments measure some perturbations from the *true* value, such that their mean could serve as a best-estimate of the *actual* precipitation value, and that other statistics (e.g., minimum, maximum, and standard deviation) could be used to estimate confidence intervals. (Note that because the number of available instruments is typically limited, the traditional 10[th] and 90[th] percentiles of a quantity as confidence intervals are of little meaning in this case). This approach serves as the basis for PrecipBE, ARM's best-estimate precipitation data product, described and demonstrated in the section below.

## 3 The PrecipBE Algorithm

The PrecipBE VAP processing is performed on a per-precipitation-event basis, leveraging ARM measurement capabilities, depending on instrument data availability per deployment, while considering QC samples and ARM data quality reports (DQRs). The VAP currently only synthesizes rainfall data (with future expansion to solid precipitation); its processing workflow is described in the flowchart shown in Figure 8. Processing begins separately for each instrument. However, because given precipitation events can persist for more than a day or through 23:59:59 UTC of a given day, data from all available instruments are loaded for up to 7 days following the currently processed day, depending on whether a continuing event is indicated by one or more of the available instruments. This buffer data loading prevents precipitation event biases driven by day-transition artifacts. Consistent with the inter-comparison discussed above, a continuing event suspect is identified if precipitation instances (precipitation amount sample greater than 0 mm) are detected by a given instrument less than 30 minutes from the end of the given day, i.e., after 23:30 UTC.

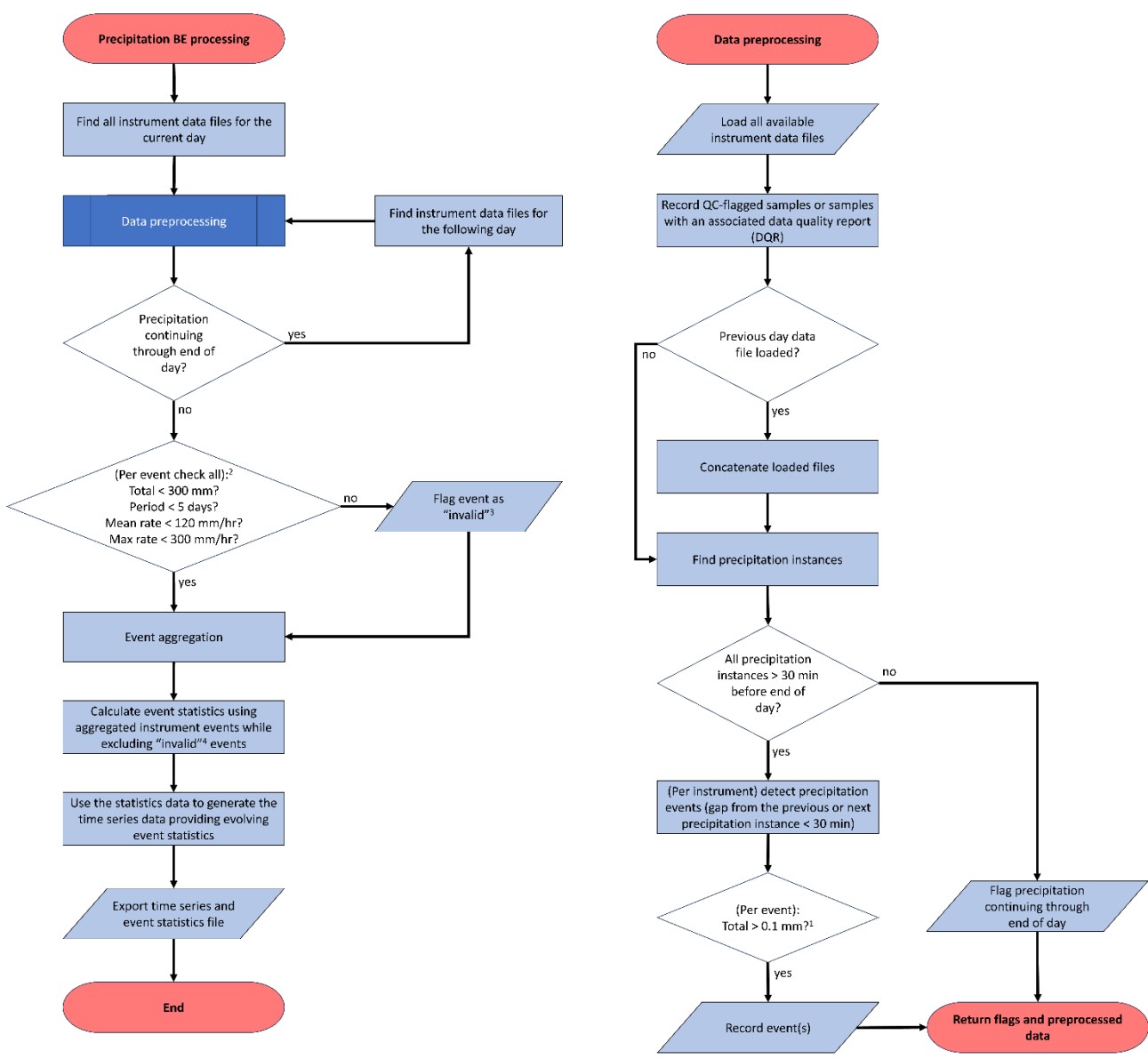

¹ 1.5 mm threshold for TBRG due to a large least count of 0.254 mm (0.01 inch) and a tendency for sporadic counts

² Event extremes can be modified or adapted for "extreme" deployment sites

³ Even with a single invalid sample, the entire event is flagged as a precaution against other bad samples that could pass the extreme threshold test

⁴ "Invalid" in this context is an event with sample(s) exceeding the threshold above, sample(s) not passing DQ tests, and/or has an active DQR associated with part or the entire event period.

**Figure 8: PrecipBE processing flowchart**

Precipitation event processing is generally similar to the methodology discussed in Section 2.1 until the event aggregation step of the flowchart. Following the inter-comparison results, the 0.1 mm cumulative precipitation event minimum is applied to all instruments, except for the TBRG, in which case a 1.5 mm threshold is used due to its coarse measurement (equivalent to a

minimum effective error of ~8.5%), and because it is more prone to sporadic counts (not shown). In addition, given the RAINWB and ORG biases demonstrated above, those two instruments are entirely omitted from the PrecipBE algorithm (see Table 1).

During the aggregation stage, all valid instrument events are aggregated together while following the 30-min no-precipitation logic discussed above. As such, if during the aggregation stage, a continuing event suspect (precipitation instances after 23:30 UTC by one or more instruments) is ultimately gapped by more than 30 min from the closest precipitation instance(s) during the following day, the event is not a continuing event, and the loaded buffer day data are discarded.

The PrecipBE algorithm robustly addresses potential issues stemming from problematic data. Here, flagged events (events with one or more QC samples or anomalous readings) or events with associated DQRs are not omitted before aggregation, as in the comparison above. Instead, all events detected by a given instrument are still included in the aggregation stage to resolve a PrecipBE event but are excluded from the PrecipBE event statistics calculations if one or more of them have one or more problematic samples. For example, in the case of the diagram shown in Figure 1, the commonly occurring interlaced event configuration will end in a single resolved PrecipBE event incorporating all four instruments (PWD, LDIS, VDIS, and TBRG), regardless of whether one or more instrument events have problematic samples or an associated DQR. Assuming that all instrument events are valid, all four of them will be included in the statistics calculation. However, assuming an issue with TBRG event 3, for example, all TBRG events will be removed from the resolved PrecipBE event statistics, which will only incorporate three instruments (PWD, LDIS, and VDIS). Assuming instead that LDIS event 4 has problematic samples, PrecipBE will still resolve a single event, even if the period between the end of VDIS event 1 and the onset of PWD event 3 is greater than 30 min. In that case, statistics will be based on the PWD, VDIS, and TBRG events. We note that other approaches, such as omitting those problematic events from the aggregation stage as well, were extensively tested and resulted in significant PrecipBE event biases driven by the sporadic nature of anomalous samples across instruments (not shown). The currently implemented approach, therefore, prevents event onset and ending inconsistency issues at the expense of fewer incorporated instruments. This approach also served as the main incentive for excluding the RAINWB instrument from the algorithm due to its substantial positive event period biases (see Figure 3e).

As suggested by the flowchart in Figure 8, once the resolved PrecipBE event statistics are calculated, they are used to generate time series data, followed by the export of daily PrecipBE files, which are described below.

## 4 PrecipBE Dataset Structure and SGP Output Demonstration

PrecipBE includes two datastreams (data set types) streamlining both process understanding and model evaluation studies using ARM surface precipitation data. The first datastream provides time series (evolving) precipitation data at 1-min temporal resolution, whereas the second includes per-event statistics in an easy-to-use one-dimensional (tabular) format. (PrecipBE data file structure and the utilization of each of these datastreams is demonstrated in a Juypter notebook available on the ARM

The time series datastream (`precipbetseries`) provides the temporally-evolving instrument-mean, minimum, maximum, and standard deviation of event-cumulative precipitation and 1-min precipitation rates. Each timestamp indicates the number of instruments used, and flags are provided for events detected using only a single instrument. The time series files also include bitwise flag arrays for instrument availability, invalid instrument samples, and instrument DQRs. Figure 9 shows an example of the PrecipBE time series output for two events that started at the SGP site on November 8, 2024, with the second event ending just after 04:00 UTC of the following day. Note that the cumulative precipitation (top panel) zeros out after the end of the first event until the beginning of the second event, enabling straightforward, low overhead, analysis. For example, in the first depicted event, cumulative precipitation increases at a varying rate with a short burst around 09:45 UTC, during which a 1-min averaged precipitation rate exceeding 200 mm/h is observed by one of the instruments (lower panel), with very weak and intermittent precipitation in the final 4.5 hours of this event.

The PrecipBE time series data suggest that none of the 7 available instruments were omitted from the statistics calculations of these two events due to flags, bogus samples, or existing DQRs. The time series data file provides information about which instruments were available via its bitwise `available_instruments` field — in this case, the SGP C1 facility's VDISQUANTS and LDQUANTS VAPs, DISDROMETER, and the WBPLUVIO2, and the SGP E13 facility's PWD, AOSMET, and TBRG. However, while this datastream provides all available precipitation data converted to accumulated totals in 1-min increments (in units of mm/min), examining statistics of particular events, such as the two depicted in Figure 9 would require additional processing. Alternatively, one could use the PrecipBE statistics datastream (`precipbestats`) files, which are only generated for days with precipitation event onsets, having the number of timestamps equal to the number of precipitation events that started on a given day. In case of November 8, 2024, illustrated below, the corresponding statistics data file includes two timestamps.

In each timestamp, `precipbestats` informs about statistics of the given event such as the instrument-mean, minimum, maximum, and standard deviation of onset, end time, period, total amount, mean precipitation rate, 1-min-averaged maximum precipitation rate, and precipitation rate standard deviation, as well as various flags and information such as which instrument recorded the highest precipitation rate or smallest total amount for that event. For example, the major precipitation event depicted in Figure 9 resulted in a cumulative amount of ~54 ± 9 mm with an instrument minimum and maximum of 43 and 75 mm, respectively. The statistics data file indicates that the DISDROMETER recorded the maximum precipitation rate of 218 mm/h during that event. In comparison, the instrument-mean maximum precipitation rate was more moderate yet still rather intense at 85 mm/h. Finally, the statistics dataset contains statistical information about the surface-level atmospheric state during precipitation events, with data harvested from (in order of preference) the MET, the automatic weather station (MAWS; Holdridge and Kyrouac, 2017), or one of the Vaisala WXT systems operated by ARM (see Table 1), as well as drop distribution moment data derived using the VDISQUANTS or LDQUANTS, depending on availability. For example, the surface temperature during the major November 8, 2024, event ranged between 9.3° and 13.6° C, with an average of 10.0° C, while

the event-mean RH was 97.5%. The even-mean liquid water content derived by the VDISQUANTS VAP was 0.3 g/m³, and the average mass-weighted mean drop diameter was ~1.5 mm.

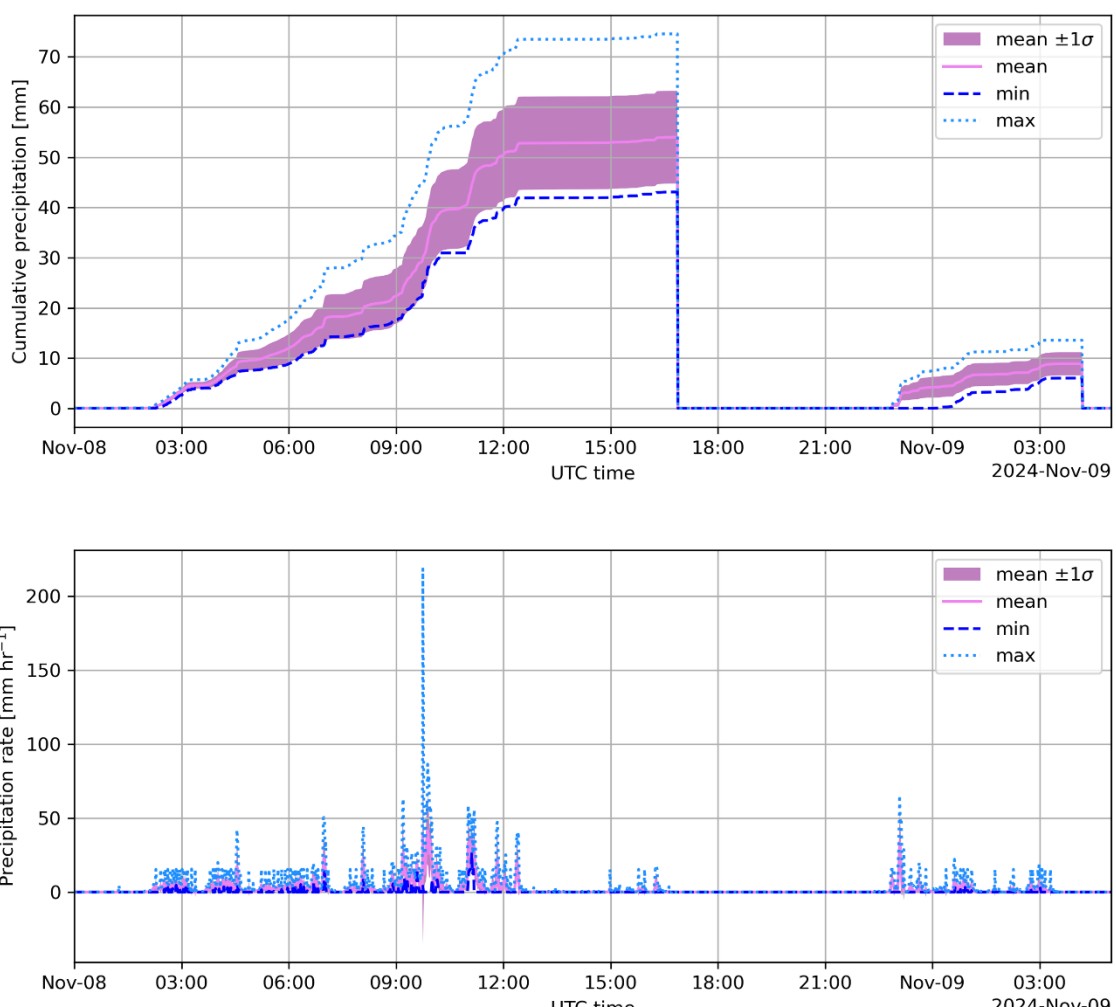

 **Figure 9: PrecipBE time series of two precipitation events that occurred on November 8, 2024, at the ARM SGP site, the second of which continued into November 9. (Top) Per-event cumulative precipitation, (bottom) precipitation rates. The plot illustrates the instrument-minimum, maximum, mean, and mean ± standard deviation ($\sigma$) (see legend).**

# 5 Long-term Trend Analysis of PrecipBE Output for the ARM SGP Site

Using PrecipBE statistics data files generated for the SGP site, spanning September 2, 1993, to March 4, 2025 (~31.5 years), we can easily examine precipitation event trends at the ARM site. Figure 10 shows running-mean time series data that facilitates basic trend analysis. We depict both curves calculated using the full dataset and curves calculated using a data subset derived only from multi-instrument events. Ideally, one should be inclined to use precipitation event properties and statistics derived from more than one instrument, as they are considered more robust than those based on a single instrument. However,

ARM operated only the TBRG starting in September, 1993, ~14 months after the SGP site launch, until April, 2006, when the DISDROMETER was deployed as the first addition to the growing suite of precipitation instruments ARM operates at the site. The results of the instrument inter-comparison in Section 2 indicated that the TBRG is generally consistent with other advanced precipitation instruments in event totals. It is also consistent with other instruments in event precipitation rates, as long as it is conditioned for event total greater than its least count by some factor (e.g., effective uncertainties of 12.7% and 8.5% at event

total of 1 and 1.5 mm, respectively). We follow these conclusions to derive the statistics depicted in Figure 10, which are also part of the motivation to examine 1-year-windows.

The 1-year running sum (annual) precipitation record (Figure 10a) largely shows little difference in annual amount between the full dataset and multi-instrument subset, with annual means of ~800 and ~750 mm (respectively) in agreement with previous studies (cf. Sisterson et al., 2016). The SGP annual rainfall is quite variable, with some years in which the site

experienced significant amounts (e.g., 2008 and 2019 exceeding 1100 mm), and others when the site exhibited small amounts (e.g., below 400 mm in 2006 and 2011). Statistically significant linear fits suggest a decadal increase in annual rainfall of more than 36 mm per decade (~5%). Those positive rainfall trends are qualitatively consistent with studies that examined single-day precipitation amount trends in station data over the south-central US, where the SGP site is located (e.g., Harp and Horton, 2022; Sun et al., 2021, their Figure 2). The number of significant precipitation events, referred to here as events with totals

exceeding 1 mm, tentatively suggests a statistically significant increasing trend (Figure 10b), commensurate with ~18 min (~7%) decadal reduction in event period (not shown). Here, the higher event total amount threshold mitigates the positive (negative) bias in the number of events (event period) in the earlier years of the SGP site, when the TBRG was the only operating precipitation-measuring instrument, such that event properties are strongly influenced by the TBRG's tendencies discussed in Section 2.2. Yet, between the full dataset and multi-instrument subset during overlapping periods, a limited

positive bias is still observed in the case of the number of events (Figure 10b). Therefore, all else being equal, it is more likely that the decadal trend leans towards the multi-instrument subset, with an increasing trend in the number of events on the order of 10 more events per year per decade. Given the definition of precipitation events in PrecipBE (precipitation instances gapped by less than 30 min), these results could indicate a growing tendency to more precipitation from broken cloud systems, which could be related to observed trends and feedbacks (e.g., Goessling et al., 2025; Loeb et al., 2024; Sherwood et al., 2020; Song

et al., 2023), yet additional research using PrecipBE and other datasets is required to support this hypothesis.

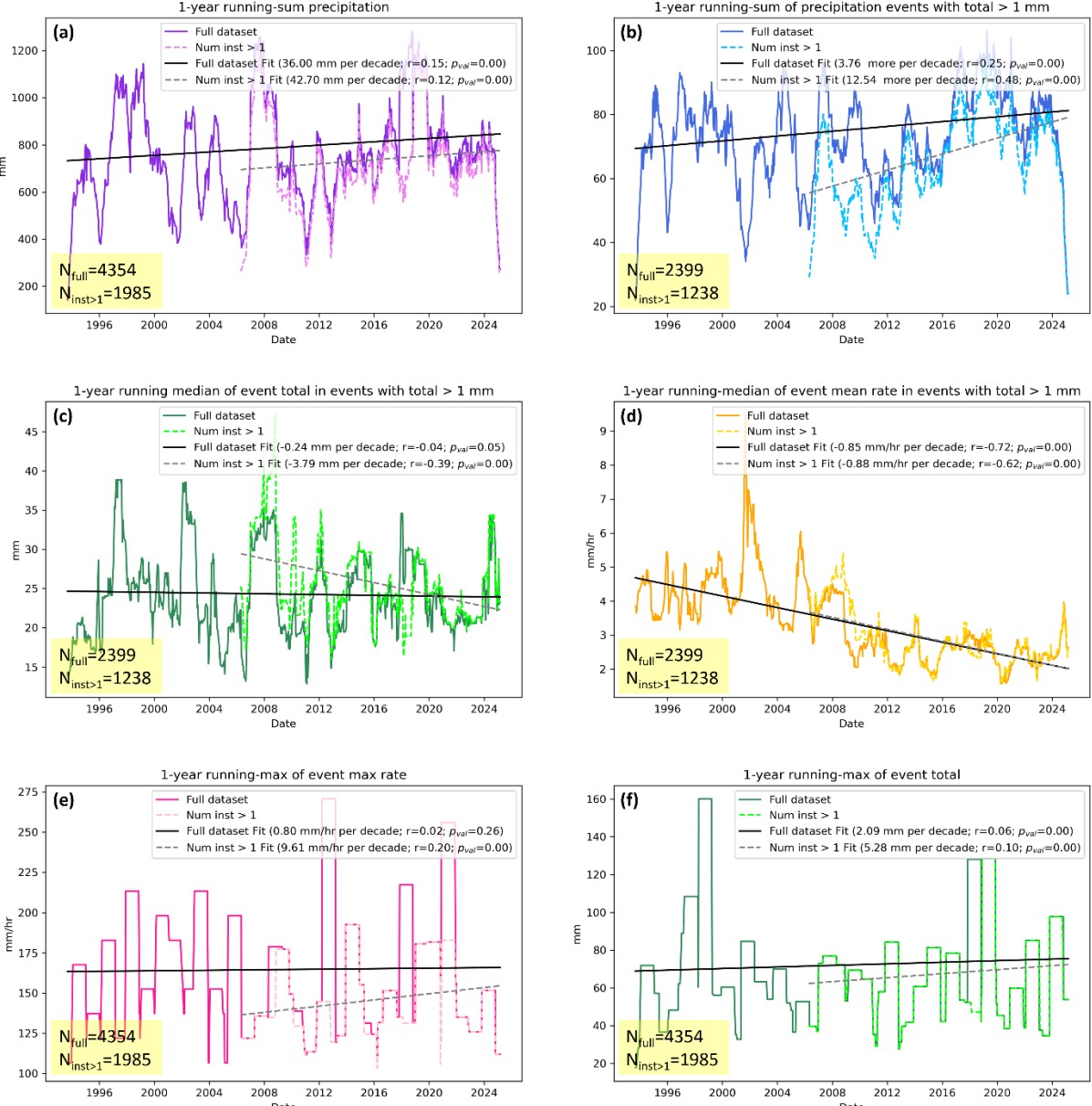

**Figure 10: Long-term trends in PrecipBE precipitation event properties for the ARM SGP site between September 2, 1993, and March 4, 2025. The solid curves were generated using all of the available precipitation event statistics, whereas the dashed curves were generated using precipitation events detected by two or more instruments (first effective sample on April 11, 2006, with the addition of the DISDROMETER). (a) 1-year running-sum (annual) precipitation totals, (b) 1-year running sum (annual) number of precipitation events with total > 1 mm, (c) 1-year running median of precipitation event total in events with total > 1 mm, (d) 1-year running median of event-mean precipitation rate in events with total > 1 mm, (e) 1-year running maximum of event 1-min-averaged maximum precipitation rate, and (f) 1-year running maximum of precipitation event total. The solid black and dashed grey lines denote linear fits to the full dataset and the multi-instrument subset, respectively. Decadal trends, correlation coefficients, and P-values are given in the legends. All quantities were calculated using the instrument-mean data. The total number of samples (precipitation events) used in the illustrated curves are given the bottom left corner of each panel.**

From a bulk perspective, all else being equal, the reduction in the precipitation event period can be translated, on average, to a trending decrease in event totals, which is indeed suggested from Figure 10c, consistent in both the full dataset and the multi-instrument subset. Following the same logic, one might expect an increasing average precipitation rate, but the 1-year running median of the event-mean precipitation rates indicates a statistically significant decreasing trend, consistent between the full and multi-instrument datasets (Figure 10d). By examining event period in the multi-instrument subset, thereby mitigating the effect of the TBRG's negative event period bias (e.g., Figure 3b), a 1-year running sum (annual) precipitation time (not shown) indicates a minimal and statistically insignificant reduction. Therefore, these results raise an apparent inconsistency between higher annual rainfall and shorter yet less intense events, on average. However, this inconsistency can be reconciled via examination of precipitation extremes in a 1-year timeframe of event maximum 1min-averaged maximum precipitation rates and maximum event totals (Figure 10e and Figure 10f, respectively). These curves exhibit general consistency between the full dataset and the multi-instrument subset and all except the maximum precipitation rate using the full dataset indicate a statistically significant increasing trend in both metrics: more than 9 mm/h per decade increase in maximum precipitation rate (6.5%) using the multi-instrument subset and between 2-5 mm per decade increase in extreme event totals (~3–7.5%) over a 1-year timeframe. Taken together, this precipitation event trend analysis indicates that the observed increase in annual rainfall could be catalyzed by a few more extreme precipitation events taking place at the SGP site. Examination of the causal sources of these trends via counterfactual exercises and their attribution to potential drivers such as regional natural variability (e.g., Higgins et al., 2007; McKinnon and Deser, 2021) or changes to the local land use (e.g., Krishnamurthy et al., 2025) remain a topic of future studies.

## 6 Conclusions and Outlook

In this study, we presented an analysis of differences in ARM precipitation instrument measurements from a unique per-event perspective. Supported by an ML application to instrument differences to examine the importance of various atmospheric state variables and parameters, the analysis indicates that, by and large, most ARM instrument rainfall observations are consistent with each other. Yet, deviations, occasionally of significant magnitudes, often occur, and could be driven by specific parameters such as relative humidity and wind properties, which could be site and deployment-dependent, or by differing instrument response functions to the same parameters those instruments are aimed at measuring (e.g., precipitation rates). Without additional prior knowledge, these results suggest that, on a first-order basis, the best estimate of precipitation properties is ostensibly that which incorporates all available valid data, which motivates the design of the PrecipBE value-added product (VAP). That said, while the analysis indicated that specific instruments show some tendency for certain behaviors, such as shorter precipitation event periods in the case of the TBRG and WBPLUVIO2, other instruments, specifically, the RAINWB and ORG exhibit clear and significant biases, which cannot be ameliorated and therefore integrated into PrecipBE. Fortunately, ARM retired the RAINWB several years ago, and the ORG is in the process of being retired in 2025.

PrecipBE provides time series and tabular statistics datasets that are easy to use and comprehensive, including precipitation event properties, and are supplemented with ancillary data from various ARM datasets. Therefore, it is likely that this VAP would become the baseline (go-to) precipitation product for the ARM user community, augmenting the derivation of scientific insights and streamlining model evaluation. Those features of this VAP were demonstrated via the examination of a single-day output as well as a long-term trend analysis of precipitation events at the ARM SGP site. The trend analysis tentatively suggests mainly shorter and less intense precipitation events at the SGP site, but also a long-term increase in annual rainfall driven primarily by more extreme event properties (event totals and maximum precipitation rates) of relatively rare, highly intense precipitation events. While we believe that numerous additional insights about surface precipitation at the SGP and other ARM sites can be derived via conditioning on various metrics related to drop size distribution moments, temperatures, diurnal cycle, time of year, etc. provided in the PrecipBE data files, we leave such analyses for the ARM user community. PrecipBE will soon become an operational product with a several-day lag from real-time, and hence, its datasets will be continuously updated and made available via the ARM Data Discovery (https://adc.arm.gov/discovery). Future planned VAP updates include the addition of solid precipitation properties at applicable sites and the potential integration of radar-based low-level precipitation estimates. We invite the ARM user community to leverage PrecipBE and provide feedback to further enhance this new and exciting data product.

**Data Availability**

Current and future releases of PrecipBE time series (Silber, 2025c, d) and statistics datasets (Silber, 2025a, b) are and will be available on the ARM Data Discovery (https://adc.arm.gov/discovery/#/results/s::precipbe). A Jupyter notebook demonstrating the structure and application of PrecipBE datasets is available on the ARM Notebooks Github repository at: https://github.com/ARM-Development/ARM-Notebooks/blob/main/VAPs/precipbe/precipbe_intro.ipynb. The precipitation datasets of the PWD (Kyrouac et al., 2021), AOSMET (Kyrouac and Tuftedal, 2010), DISDROMETER (Wang, 2006), VDISQUANTS (Hardin et al., 2021), LDQUANTS (Hardin et al., 2019), TBRG (Kyrouac et al., 2006, 2021), WBPLUVIO2 (Zhu et al., 2016), RAINWB (Shi et al., 2010), and ORG (Kyrouac et al., 2021) from the ARM SGP, ENA, and TRACER sites are available on the ARM Data Discovery (https://adc.arm.gov/discovery/; last access: 10 March 2025).

**Author Contribution**

Conceptualization: IS, JMC, and AT
Formal analysis, investigation, methodology, visualization, and original draft preparation: IS
Project administration: JMC
Data curation and validation: IS and MRK
Manuscript review and editing: all authors

## Competing Interests

The authors declare that they have no competing interests

## Acknowledgments

The authors thank Scott Collis and Scott Giangrande for valuable comments and feedback. Data were obtained from the ARM user facility, a U.S. Department of Energy (DOE) Office of Science user facility managed by the Biological and Environmental Research (BER) program.

## Financial Support

This research was supported by the ARM user facility, a U.S. Department of Energy (DOE) Office of Science user facility managed by the Biological and Environmental Research (BER) program. Pacific Northwest National Laboratory is operated for the U.S. Department of Energy by Battelle under contract DE-AC05-76RL01830.

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
