# Peer review of "Synthesis of ARM User Facility Surface Rainfall Datasets to Construct a Best Estimate Value Added Product (PrecipBE)"

_EGUsphere, 2025_

## Referee Comment (RC1)

**Review of Synthesis of ARM User Facility Surface Precipitation Datasets to Construct a Best Estimate Value Added Product (PrecipBE), by Silber et al.**

This paper presents a new ARM product that generates a best estimate (with uncertainties derived from the variability from different sources) of surface precipitation. Although I have a few comments on the paper aimed at improving it, my opinion is that the paper should be accepted for publication after addressing these comments.

**Comments:**

1. Use of precipitation throughout the manuscript. After reading the title, I was quite excited to see that this study offers a precipitation product, because rainfall is easier to extract from these sensors than melting snow or snow or hail. However, digging through the paper, it appeared quickly that the study focusses on rainfall, not all types of precipitation. First, the study only uses SGP data, which presumably encounters little solid precipitation at ground. Since the investigation required to develop a solid precipitation product would require another paper of its own, the only option seems to change "precipitation" to "rainfall" throughout the manuscript. The name of the product is also very misleading for the same reason.

2. To continue on the same subject, unless I missed it, I assume that you have screened out solid and melting precipitation with a ground-level temperature threshold (10C is what I'd do)?

3. The relevance for all precipitation regimes covered by the ARM program. I understand why you have focussed on the SGP data (multitude of sensors), however the main issue I have with this is that the precipBE product is expected to be produced over all ARM sites. If that's the case, I really wonder how the findings of this paper would change if you were to repeat the same analysis for tropical rainfall or drizzling clouds in the Sc regimes. At the very least, I think this limitation of the study needs to be acknowledged and discussed.

4. Figure 1 and associated discussion: The main issue here is that the tipping bucket type of instruments needs some accumulation before tipping. However, for the "continuous" measurements, when all instruments are working, the different onset and end times of events is part of the errors and are associated with the different sensitivities of the instruments. So I feel that by discarding these data, you are not including that error source in your analysis. I would suggest a separate analysis for the continuous measurements.

5. Figures 2,3,4,5: I have a few comments about all these figures.
   a. First you miss an x-axis on them (at the bottom).
   b. Because you are showing PDFs of differences, it is difficult for the reader unfamiliar with typical rainfall values to assess from these figures whether those biases and std are large. I would suggest adding one figure between the current figure 1 and 2 that shows the PDFs of the quantities themselves (from the PWD since that's what you use as a reference) to help assessment of figures 2 to 5. For instance, is 3-5 mm a big difference in event total amount relative to the typical total amounts? Can't tell from the material presented here.
   c. I think the analysis could be a little deeper about the differences in those plots. For instance, taking the example of event total amount again, how do these errors look like for low, intermediate or extreme event total amounts? This is a very important information to provide to downstream users. What I would suggest is split your PDF

of event total amount (as an example) in three or four (terciles or quartiles) and produce the difference PDFs for each of these terciles or quartiles.

Minor edits:

1. Table 1: should RAINCAP be defined itf it's an acronym (I don't know what that is).
2. Lines 253-255: I think that observations / suggestion is a very good reason to do the tercile or quartile analysis. We need to see a quantitative analysis of this for all quantities plotted in Figures 2 to 5.
3. Line 332: maybe add explicitly the temporal resolution of the time series (I expect it's 1 minute ?).
4. Lines 394-395 " trends are consistent with studies …". Could you briefly mention what sort of data was used in these studies and maybe what range of trends was found in these studies ?

*Good luck with the revision,*

*Alain Protat*

*Bureau of Meteorology, Melbourne, Australia*

*12/11/2025.*

---

## Author Comment (AC1)

**Author Responses**

We thank the reviewers for their valuable comments and helpful suggestions, which we think helped us improve the quality of the manuscript. Our responses and revisions are enumerated below. Beyond changes in response to reviewer comments, we split Section 2.2 into two parts (as it had become quite long) and made minor copy edits.

**Reviewer #1 (Alain Protat):**

This paper presents a new ARM product that generates a best estimate (with uncertainties derived from the variability from different sources) of surface precipitation. Although I have a few comments on the paper aimed at improving it, my opinion is that the paper should be accepted for publication after addressing these comments.

1. Use of precipitation throughout the manuscript. After reading the title, I was quite excited to see that this study offers a precipitation product, because rainfall is easier to extract from these sensors than melting snow or snow or hail. However, digging through the paper, it appeared quickly that the study focusses on rainfall, not all types of precipitation. First, the study only uses SGP data, which presumably encounters little solid precipitation at ground. Since the investigation required to develop a solid precipitation product would require another paper of its own, the only option seems to change "precipitation" to "rainfall" throughout the manuscript. The name of the product is also very misleading for the same reason.

We agree with the reviewer that the focus on rainfall rather than rainfall and snowfall was not communicated properly. We updated the text accordingly, where appropriate. This includes various text edits, such as the modified title:

*Synthesis of ARM User Facility Surface Rainfall Datasets to Construct a Best Estimate Value Added Product (PrecipBE)*

the manuscript's abstract:

*"Using a long-term instrument inter-comparison ... we demonstrate that ARM rainfall-measuring instruments are generally consistent with each other at the statistical level. Inter-instrument deviations at the single event level can be large, especially at specific rainfall event properties such as maximum precipitation rates ... While ... PrecipBE datasets ... cover rainfall from multiple ARM deployments up to March 2025, PrecipBE is planned to be expanded to include solid-phase precipitation and will soon become an operational product with a several-day lag from real-time."*

Section 3 (VAP description):

*"The PrecipBE VAP ... currently only synthesizes rainfall data (with future expansion to solid precipitation); its processing workflow is described in the flowchart shown in Figure 6."*

and Section 6 (conclusions; text not modified):

*"... the analysis indicates that, by and large, most ARM instrument rainfall observations are consistent with each other... Future planned VAP updates include the addition of solid precipitation properties at applicable sites and ..."*

The product's name (PrecipBE) is intentional, as we plan to integrate solid-phase precipitation data. The required machinery for this integration is fully developed in the PrecipBE codebase. The main bottleneck here is the refinement of snowfall observations and retrievals, a challenge of its

own, as the reviewer has noted. We are working on this (mainly using the ARM North Slope of Alaska site observations), but this could take some time to reach a satisfactory level in the various snow-measuring instruments ARM operates. Given the pressing need of ARM users for a BE product that integrates the full suite of rainfall measurements, we decided to initiate the process by releasing the liquid precipitation component first.

2. To continue on the same subject, unless I missed it, I assume that you have screened out solid and melting precipitation with a ground-level temperature threshold (10C is what I'd do)?

Thank you for raising this point. Events with an average 2 m temperature < 3° C are flagged in PrecipBE. We agree that 10 °C is an unerring option, but from our experience, 3 °C effectively addresses that issue, at least in currently available ARM deployments, without omitting a significant fraction of events. For comparison, precipitation events with an average 2 m temperature < 3° C account for ~15% of the examined SGP dataset, whereas those with an average 2 m temperature < 10° C account for ~32%. In any case, after revisiting the inter-comparison code, it appears that we eventually used plots that do not exclude these temperature < 3° C events. Figures 2-5 were updated, and the text was modified accordingly. We now clarify the temperature constraint in the text (Section 2.1):

*"For a given instrument, we define a rainfall event as a set of positive accumulated precipitation samples (at temperatures greater than 3 °C)… ",*

and note that while the quantitative results slightly changed, our qualitative conclusions did not.

3. The relevance for all precipitation regimes covered by the ARM program. I understand why you have focussed on the SGP data (multitude of sensors), however the main issue I have with this is that the precipBE product is expected to be produced over all ARM sites. If that's the case, I really wonder how the findings of this paper would change if you were to repeat the same analysis for tropical rainfall or drizzling clouds in the Sc regimes. At the very least, I think this limitation of the study needs to be acknowledged and discussed.

That is a good point. We actually ran this inter-comparison across a large number of ARM sites and, generally speaking, received qualitatively similar bulk results. (We note in some cases, a specific instrument exhibited a clear bias, which was informative). We adopted the reviewer's suggestion to add analysis results for convective and stratiform sites (represented by TRACER and ENA, respectively), which are presented in the newly added figures 6 and 7 and discussed as part of the ML analysis results in Sect. 2.3. (For brevity, we focus only on the maximum precipitation rate rather than all four event properties.) We think these figures provide stronger support for our results.

4. Figure 1 and associated discussion: The main issue here is that the tipping bucket type of instruments needs some accumulation before tipping. However, for the "continuous" measurements, when all instruments are working, the different onset and end times of events is part of the errors and are associated with the different sensitivities of the instruments. So I feel that by discarding these data, you are not including that error source in your analysis. I would suggest a separate analysis for the continuous measurements.

We agree that there's a challenge in evaluating continuous vs. more discrete observations (from the rain gauges). That was part of the motivation for using instrument pairs in this analysis rather than integrating all instruments in the first place, as well as for taking the event aggregation approach described in the methodology section, using Figure 1 for support. Note that this would

have become a more significant issue if the TBRG was serving as the reference instrument, but because we used the "continuous-type" PWD as reference, the TBRG quantization resulted in robust event aggregation and minimal removal of TBRG events, as already noted in the text:

"*This event association and aggregation exercise results in the removal of some instrument pair events. Removal percentages range from 0.7% of TBRG events …* "

The large least count of the TBRG is also already discussed in the text, for example:

"*Ideally, the best reference instrument would be the tipping bucket rain gauge (TBRG) … However, the TBRG has a very coarse precipitation amount least count (minimum detection of 0.254 mm; 0.1 inch; cf. Table 1), rendering its sensitivity and general accuracy (in weak events) inadequate for serving as a reference instrument (as demonstrated below), especially compared to other instruments such as disdrometers.*"

Another example (note the reference to the reviewer's tercile suggestion below):

"*In both the mean and 1-min maximum precipitation rates, the TBRG (Figure 4b and Figure 5b) exhibits a distinct bi-modal PDF shape, which originates in its coarse minimum least count of 0.254 mm. The events associated with the secondary peak in the mean precipitation rate histogram (Figure 5b) are at the third PWD distribution tercile (yellow-shaded area), i.e., intense enough to be detected by the TBRG, but too weak and/or short to form consistent correspondence with the other instruments, and possibly influenced by some residual water on the bucket's "spoon". The coarse TBRG least count, combined with the 1-min sampling resolution, also results in weak events being below the TBRG's detection limit, as evident by the lack of first tercile events (purple-shaded areas) based on the maximum precipitation rate (Figure 4b) and very few weak events when partitioned based on mean event precipitation rate (Figure 5b). Accounting for this instrument limitation by omitting precipitation events with total amounts smaller than 1 mm results in a behavior consistent with the above-mentioned instruments and the disappearance of the bi-modal PDF artifact (not shown), suggesting that higher event total thresholds should be used for the TBRG in an integrated data product.*"

The large least count also motivated implementing a larger minimum event total threshold for integration in PrecipBE to mitigate associated biases as already noted in the text:

"*Following the inter-comparison results, the 0.1 mm cumulative precipitation event minimum is applied to all instruments, except for the TBRG, in which case a 1.5 mm threshold is used due to its coarse measurement (equivalent to a minimum effective error of ~8.5%), and because it is more prone to sporadic counts (not shown).* "

As mentioned in the text and quoted above, the analysis using a minimum event total threshold of 1 mm, resolved some of the noticeable issues such as the precipitation rate bi-modality driven by this coarse discretization of the TBRG, but our qualitative conclusions still hold. For reference, below is an example from the 1 mm threshold analysis shown for mean precipitation rates (note the PWD's distribution in the top panel and tercile depiction in response to the reviewer's comment below):

[Figure]

5. Figures 2,3,4,5: I have a few comments about all these figures.

a. First you miss an x-axis on them (at the bottom).

Fixed in Figures 2-5 (see example above).

b. Because you are showing PDFs of differences, it is difficult for the reader unfamiliar with typical rainfall values to assess from these figures whether those biases and std are large. I would suggest adding one figure between the current figure 1 and 2 that shows the PDFs of the quantities themselves (from the PWD since that's what you use as a reference) to help assessment of figures 2 to 5. For instance, is 3-5 mm a big difference in event total amount relative to the typical total amounts? Can't tell from the material presented here.

Good point. The reviewer's suggestion was implemented in the top panel of figs. 2-5 as demonstrated above.

c. I think the analysis could be a little deeper about the differences in those plots. For instance, taking the example of event total amount again, how do these errors look like for low, intermediate or extreme event total amounts? This is a very important information to provide to downstream users. What I would suggest is split your PDF of event total amount (as an example) in three or four (terciles or quartiles) and produce the difference PDFs for each of these terciles or quartiles.

Excellent suggestion. We partitioned the histograms based on the PWD distribution's terciles (as demonstrated above) and refer to the different ranges throughout the text. For example, see our response to comment #4 above, or:

*"Similar to the event total, event periods are positively skewed (Figure 3a), averaging at 105 min, just above the second tercile… The DISDROMETER shows a greater tendency, with precipitation events lasting 120 min longer on average, and a trend toward extreme values in cases within the third tercile (yellow-shaded areas in Figure 3h)."*

Table 1: should RAINCAP be defined itf it's an acronym (I don't know what that is).

RAINCAP is not an acronym but simply a trademark name by Vaisala Inc.

Lines 253-255: I think that observations / suggestion is a very good reason to do the tercile or quartile analysis. We need to see a quantitative analysis of this for all quantities plotted in Figures 2 to 5.

Right. See our response above and the updated figures. We hope this would address the reviewer's concern.

Line 332: maybe add explicitly the temporal resolution of the time series (I expect it's 1 minute ?).

Added:

*"The first datastream provides time series (evolving) precipitation data at 1-min temporal resolution, whereas the second includes per-event statistics in an easy-to-use one-dimensional (tabular) format."*

Lines 394-395 " trends are consistent with studies …". Could you briefly mention what sort of data was used in these studies and maybe what range of trends was found in these studies ?

Those two referenced studies were based on NOAA's Global Historical Climatology Network Daily (GHCN-D) datasets using station data. Their analyses were based on daily precipitation data; therefore, a quantitative comparison is prone to bias. We now clarify in the text that the agreement is qualitative and direct the reader to a specific figure in the second reference:

*"Statistically significant linear fits suggest a decadal increase in annual rainfall of more than 36 mm per decade (~5%). Those positive rainfall trends are qualitatively consistent with studies that*

*examined single-day precipitation amount trends in station data over the south-central US, where the SGP site is located (e.g., Harp and Horton, 2022; Sun et al., 2021, their Figure 2)."*

**Reviewer #2 (David Dufton):**

The paper presents a thorough cross comparison of multiple precipitation sensors deployed at the ARM Southern Great Plains site over the last 30 years alongside a methodology for, description and application of a new data product which combines those sensors. The paper is well written and easy to understand. My review focuses mainly on the first cross comparison section as the new value added product, while not ground breaking in its methodology, is a sensible addition, and is well described with a fitting application chosen to use as an example.

In addition to the comments within the attached pdf I have the following more general comments which are easier to present here. (We respond below to each of the reviewer comments including those from the PDF file)

1. The authors do not give reference at any point to the WMO intercomparison studies (both field and laboratory) of rain gauges (see citations in the pdf). I believe these papers may provide the authors with extra context for their findings and can be incorporated well into the paper. In particular, they should provide more information around the choice of the PWD as the reference sensor, and the potential implications for bias this may introduce.

Thanks for suggesting the WMO intercomparison reference, which we missed in our literature survey. We agree with the reviewer that it provides good context on certain aspects (e.g., a true benchmark) and is ostensibly one of the few relevant references to our analysis of maximum precipitation rates per event (the only variable examined in that intercomparison). We refer to that study at multiple locations in the text, for example:

*"To streamline the interpretation of analysis results, we select a "reference" instrument to examine deviations of events from one instrument to another… This "reference" instrument is not a "true" benchmark, as in the case of the World Meteorological Organisation (WMO) rainfall intensity intercomparison, for example (Lanza et al., 2010; Lanza and Vuerich, 2009; Vuerich et al., 2009), during which only maximum precipitation rates per event were evaluated against a reference set of carefully calibrated rain gauges. Here, however, the related biases of the "reference" instrument can still be characterized."*

While the WMO intercomparison suggested that the PWD exhibits a clear deviation from the reference, we do not find any indication of such a distinct bias. This could be related to the specific instrument used in our analysis versus the intercomparison, site climatology (Vuerich et al. mention winds not exceeding 5 m/s while excluding foggy conditions), and/or the limited number of precipitation events examined in the intercomparison — several tens of events, about 1/10 of the number of events examined here. We also wish to emphasize that the Pluvio weighing bucket, which performed best in the WMO intercomparison, is the instrument with the best agreement with the PWD here. This agreement is even more pronounced when we condition on wind intensity below 10 m/s and minimum precipitation event totals of 1 mm, conditions that are closer to the WMO intercomparison's site and data processing. In this case, the mean deviation between the PWD and Pluvio is 0.0 m/s (see figure below).

[Figure]

We now elaborate on this comparison in the text:

*"The event 1-min-average maximum and event-mean precipitation rate comparisons (Figure 4 and Figure 5, respectively) suggest that most instruments are generally consistent with each other at the bulk level… We do not see any indications for a significant negative bias of the event maximum precipitation rate by the PWD or a positive bias by the LDQUANTS product (Parsivel2), as suggested by the WMO intercomparison (cf. Lanza et al., 2010). Single-event deviations can be quite large, as depicted by distribution tails, but from a bulk perspective, the instrument pair deviations generally tend to be evenly distributed around 0 mm/h in most cases, and the PWD is*

*most consistent with the LDQUANTS (Figure 4f) and WBPLUVIO2 (Figure 4i). The agreement with the WBPLUVIO2 averages at 0.0 mm/h when conditioning on event totals greater than 1 mm and wind speed smaller than 10 m/s (not shown), close to the operation and filtering conditions of the WMO intercomparison (see Lanza and Vuerich, 2009; Vuerich et al., 2009; sect. 3.1). This result contrasts with the WMO intercomparison, where the Pluvio exhibited the highest performance (Lanza et al., 2010; their Table 2). This contrast is potentially influenced by deployment setup, site-specific factors, and/or sample size (the WMO intercomparison used approximately 1/10 the number of precipitation events analyzed here)."*

2. The random forest application add little value to the paper. Only 2 of the models have an R2 above 0.5 and in both those cases the RF is dominated by the event mean and max rate. I'd suggest removing it, especially as its findings don't impact the resulting development of the new combined product.

We understand the reviewer's concern. Note that the $R^2$ metric used here is a multivariate one, and its main purpose is to evaluate the Random Forest fit. (Higher $R^2$ is better, and 0.5 is certainly a good threshold to work with, but for our purposes, at the very least, the value must be above 0 to suggest any predictive capabilities.) We think that the RF analysis is critical for communicating the impact of instrument sensitivity, deployment configuration, and atmospheric state on measurements (also in relation to the WMO comparison), and for the fact that the entanglement of these effects cannot be fully predicted in advance. Therefore, to have better flow in the text, now that we give relatively more attention to the maximum precipitation rates, we performed the following edits to the text:

1. Partitioned section 2.2 into two, with the second part (now section 2.3) focusing on the RF analysis.
2. Refocused the RF analysis on the maximum precipitation rates, which is typically accompanied by higher $R^2$, thereby providing stronger support for the discussed effects and the challenges in predicting them.
3. Expanded the RF analysis by incorporating equivalent output using data from a convective-dominated site (TRACER) and a stratiform-dominated site (ENA).

3. The methodology chosen for the combined methodology does not take into account the findings from the previous section beyond the removal of the two gauges that deviate most strongly. I believe if the WMO intercomparison results are considered and the findings in section 2 there is potential future scope for an update which weights the instruments based on previously published accuracy findings and their accuracy within each event (based on duration and total accumulation). For instance the TBR and Pluvio would seem to be the more accurate choices based on the WMO results, and the mean could be weighted to account for this in events which suit their use. I'd suggest the authors consider this as a future extension/improvement rather than discounting the current methodology which provides a sensible first step.

We agree with the reviewer that ultimately, weighing different instruments would ostensibly result in the best estimate of precipitation properties, and future site-specific dedicated efforts might do just that. As we discuss in the text (mainly in the RF analysis), the entanglement of the above-mentioned effects cannot be fully predicted in advance; hence, weighing based on the (hitherto limited and condition-specific) literature is prone to bias, accurate as a given intercomparison may be. This is the conundrum we mentioned in the text: we want the best-estimate product, but for such a product, we first need to characterize processed data, which would ideally be synthesized

using a best-estimate product. We added and edited the text at the end of section 2.3 to "bring this home":

*"A tentative conclusion that can be drawn from these [RF analysis] results is that weighing different instruments based on their evaluated sensitivities and accuracies from the literature can result in greater bias due to unmatching background conditions as well as unanticipated confounding factors, particularly when combining climatological factors with specific deployment setups. While additional quantitative characterization of instrument susceptibilities to deployment properties and conditions is essential, a deployment-dependent study of this type requires significant effort. The outcomes of such extensive efforts are highly challenging to predict in advance. Therefore, these data characterization studies often take place post-deployment, when the collected dataset is sufficiently large to produce substantial results (beyond the scope of this study).*

*Suppose one wishes to develop an operational, unbiased (or at least, bias-mitigated) precipitation best-estimate data product. In that case, given that they do not have a true benchmark, they need to be aware of all the factors described above by performing robust characterization, which would ideally require a best-estimate product — this presents a conundrum. A first step towards resolving this conundrum would be to assume, given the evidence from this inter-comparison about instrument consistency, that the suite of ARM instruments measure some perturbations from the true value, such that their mean could serve as a best-estimate of the actual precipitation value, and that other statistics (e.g., minimum, maximum, and standard deviation) could be used to estimate confidence intervals. (Note that because the number of available instruments is typically limited, the traditional 10th and 90th percentiles of a quantity as confidence intervals are of little meaning in this case). This approach serves as the basis for PrecipBE, ARM's best-estimate precipitation data product described and demonstrated in the section below. "*

Comments from the PDF file:

l. 64 - No reference to the WMO field and lab studies. Also given these findings their is no description here of the actual siting of the instruments across the site and how that may impact the results

Reference to the WMO intercomparison was added to the beginning of that paragraph.

l. 69 - Add a location map, plus locations of the insturments around the site.

We now refer (at the beginning of section 2.1) to a link on ARM.gov where readers can find information about the site as well as the instrument layout:

*"The analysis focuses on precipitation rainfall data collected at the ARM SGP site's co-located central (C1) and extended facility 13 (E13) over a 14-year period, from January 10, 2011, to January 10, 2025. A list of the instruments and data products analyzed is provided in Table 1. (Refer to https://armgov.svcs.arm.gov/capabilities/observatories/sgp for site information and central facility layout). "*

l. 70 "machine learning (ML) algorithm") - Be specific here

Text modified accordingly:

*"Supported by the application of a machine learning (ML) algorithm (a random forest regressor), ..."*

l. 77 – Section 5 → Section 6

Fixed. Thank you!

l. 88 - See earlier comment, a map would help.

See our response above.

l. 90 – Probably a silly question, do any of the instruments give negative accumulations? If not, I'd remove the word positive as unnecessary.

Good point. Removed.

l. 99 - "During ARM deployments" - Are these instruments mobile? Can you make this clear.

ARM operates three fixed sites (observatories) and three mobile facilities that are typically deployed for a minimum period of several months (see https://armgov.svcs.arm.gov/capabilities/observatories)

l. 100 – "up to a few percent of precipitation events" - Can you be precise. How many events are excluded due to these constraints? What would be the effect of including them?

Added: "< 2.5%" to that sentence. Those bogus instrument readings (affecting up to 33 events out of a maximum 1,330 events — aggregation-dependent) could originate from various instrument operation mechanism-dependent sources. Since we know that they are necessarily bad, incorporating them could have biased our results. To keep the text concise, we prefer to exclude that extra information, which could be somewhat confusing and disrupt the flow.

l. 104-107 – "datasets of the PWD …" - This text doesn't follow on well, the paragraph appears to be missing a start. I'm not sure where this should fit in, but it needs fixing.

Good catch. We accidentally had here part of the Data Availability Statement (now removed).

Table 1 - Add more detail to this table. Particularly the measuring principle, the sensitivities of the instruments, the quantitization intervals, and the temporal resolution. Siting constraints would be useful too, including presence or absence of wind shields, height above surface etc.

We added the temporal resolution and effective quantization increments (reported precipitation variable's increments converted to mm/min) to Table 1. The measuring principle is provided in the instrument description and instrument sensitivities are provided (occasionally in tables) in each of the references. Siting constraints vary by site, and part of the purpose of this table is to provide general information relevant to PrecipBE, so we do not wish to include site-specific information here.

l. 125 – "Present weather detector (PWD) - Lanza, L.G., Vuerich, E., Gnecco, I., 2010. Analysis of highly accurate rain intensity measurements from a field test site, in: Advances in Geosciences. Presented at the Precipitation: Measurement, Climatology, Remote Sensing, and Modeling (EGU Session 2009) - EGU General Assembly 2009, Vienna, Austria, 19–24 April 2009, Copernicus GmbH, pp. 37–44. https://doi.org/10.5194/adgeo-25-37-2010

Tendancy to always overestimate. May not be the best choice in terms of accuracy, but good in terms of data length so I can appreciate why it was chosen. Might be worth referencing these findings though.

Done. See our response to major comment #1 above.

Figure 1 - This would be better illustrated (or complemented) by a real-world example of 4 timeseries, indicating the events for each instrument. This would give a real-world example to underpin the approach, and likely contributing factors in the context of instrument resolution and siting within the SGP.

We respectfully disagree. From our experience, prior presentations of this part, which is somewhat confusing in the first place, resulted in more confusion when we used real examples. We agree that it is possible that for some readers, a real time series might be easier to comprehend, but we suspect (though we could be wrong) that for the majority, a conceptual diagram is easier to understand and therefore prefer to retain the current formatting.

l. 150 – "Bartholomeew, 2019" - All these references are also in the table, I think they can be removed here to make the text easier to read.

Done.

l. 154-156 – "We note that the filtering of QC-flagged or anomalous reading events prior to the aggregation exercise had minor influence on analysis results (not shown), but it could theoretically be more impactful in other cases." - I'm still not certain, as yet, why the above paragraph is so. I can understand TBR having shorter events or multiple events within one PWD event due to its resolution but am uncertain about the others. Having a map of the site may help, and adding the info in table 1 too. Is it due to fog impacting the optical instruments? If you then set a limit to the lowest observable 1 minute rainfall that is consistent across all instruments excluding the TBR would this change the number of events that are lost by breaking up VDIS events for example?

We think we understand the reviewer's concern. This is simply because such QC-flagged reading events were few and far between and often did not occur during a real precipitation event. Moreover, in the vast majority of QC-flagged periods, there were simply no accumulation readings → no precipitation event → exclusion from the analysis in the first place. Therefore, we noted in the text that theoretically, they could be more impactful in other sites. We would like to stress that this issue is considered in PrecipBE processing, as we already mention in the text:

*"The PrecipBE algorithm robustly addresses potential issues stemming from problematic data. Here, flagged events (events with one or more QC samples or anomalous readings) or events with associated DQRs are not omitted before aggregation, as in the comparison above. Instead, all events detected by a given instrument are still included in the aggregation stage to resolve a PrecipBE event but are excluded from the PrecipBE event statistics calculations if one or more of them have one or more problematic samples."*

Figure 2 - Add x-axis units directly. Could consider using a log scale for the y-axes, which would highlight the tails of the distributions more for comparison. All the panels have a clearly defined central peak around zero, though the extra lines make it harder to see the precise balance between positive and negative bias for each case. Instead of lines could the alternates use open histograms instead to make it easier to see the bars, especially in the tails?

x-axis units were added. We tried using a log scale for the y-axis, as shown below for the TBRG event total difference, but we think that the result is more misleading than informative. Instead, as shown in the example above in response to major comment #1, we changed the color palette, which alleviates the problem of differentiation between the RH curve and histogram bars.

[Figure]

L. 204 – " Accounting for this instrument limitation by omitting precipitation events with total amounts smaller than 1 mm results in a behavior consistent with the above-mentioned instruments…" - This is reassuring.

We agree. Refer to our response to major comment #1 for an example of that 1 mm evaluation.